# Deacetylation induced nuclear condensation of HP1γ promotes multiple myeloma drug resistance

Xin Li[1], Sheng Wang[1], Ying Xie[1], Hongmei Jiang[1], Jing Guo[1], Yixuan Wang[1], Ziyi Peng[1], Meilin Hu[2], Mengqi Wang[1], Jingya Wang[1], Qian Li[3], Yafei Wang[3] & Zhiqiang Liu ☉[1] ✉

Acquired chemoresistance to proteasome inhibitors is a major obstacle in managing multiple myeloma but key regulators and underlying mechanisms still remain to be explored. We find that high level of HP1γ is associated with low acetylation modification in the bortezomib-resistant myeloma cells using SILAC-based acetyl-proteomics assay, and higher HP1γ level is positively correlated with poorer outcomes in the clinic. Mechanistically, elevated HDAC1 in the bortezomib-resistant myeloma cells deacetylates HP1γ at lysine 5 and consequently alleviates the ubiquitin-mediated protein degradation, as well as the aberrant DNA repair capacity. HP1γ interacts with the MDC1 to induce DNA repair, and simultaneously the deacetylation modification and the interaction with MDC1 enhance the nuclear condensation of HP1γ protein and the chromatin accessibility of its target genes governing sensitivity to proteasome inhibitors, such as *CD40, FOS* and *JUN*. Thus, targeting HP1γ stability by using HDAC1 inhibitor re-sensitizes bortezomib-resistant myeloma cells to proteasome inhibitors treatment in vitro and in vivo. Our findings elucidate a previously unrecognized role of HP1γ in inducing drug resistance to proteasome inhibitors of myeloma cells and suggest that targeting HP1γ may be efficacious for overcoming drug resistance in refractory or relapsed multiple myeloma patients.

Eventually developed chemoresistance to proteasome inhibitors (PIs) is a major obstacle to the successful management of multiple myeloma (MM)[1]. Defining the key prognostic markers representing the precise molecular heterogeneity and molecular mechanisms of drug resistance to PIs will benefit designs of more effective combination therapies. Genetic instability and abnormalities are two hallmarks of MM cells, and aberrant DNA repair plays critical roles in the progression of MM and treatment response in the clinic[2]. Targeting DNA repair pathways either through non-coding RNAs or RNA splicing regulators could induce apoptosis and sensitize MM cells to the current treatment regimen[3,4]. Moreover, PIs impair the homologous recombination (HR) in MM, resulting in contextual synthetic lethality when combined with PARP inhibitors[5]. Nevertheless, the large-scale underlying mechanisms are still unclear, and a better knowledge of DNA repair pathways in MM may help optimize current regimens, thus improving the efficacy of disease treatment.

[1]The Province and Ministry Co-sponsored Collaborative Innovation Center for Medical Epigenetics; Tianjin Key Laboratory of Cellular Homeostasis and Human Diseases; Department of Physiology and Pathophysiology, School of Basic Medical Science, Tianjin Medical University, Heping, Tianjin 300070, China. [2]Tianjin Medical University School of Stomatology, Tianjin Medical University, Heping, Tianjin 300070, China. [3]Department of Hematology, Tianjin Medical University Cancer Institute and Hospital, National Clinical Research Center for Cancer, Key Laboratory of Cancer Prevention and Therapy, Tianjin's Clinical Research Center for Cancer, Tianjin 300060, China. ✉e-mail: zhiqiangliu@tmu.edu.cn

The heterochromatin protein 1 (HP1) family consists of HP1α, HP1β, and HP1γ, which contain a chromodomain (CD) and chromoshadow domain (CSD)[6]. As readers of histone 3 lysine 9 di/trimethylations (H3K9me2/3), HP1 proteins play important roles in transcriptional regulation, chromatin remodeling, cell cycle regulation, DNA replication and repair[7–9]. HP1α and HP1β are distributed in constitutive heterochromatin such as centrosomes and telomeres, and HP1γ is distributed in both euchromatin and heterochromatin. Functionally, HP1γ but not HP1α or HP1β is associated with actively transcribed gene regions and plays a role in efficient transcriptional elongation[10]. As a reader of H3K9me2/3[11], HP1γ also serves as a binding platform for the recruitment of other histone modifiers, thus to form a "histone code" in gene expression at a "on-the-gene" pattern[12]. It was recently shown that HP1α protein forms liquid-liquid phase separation (LLPS) upon binding DNA, indicating that heterochromatin-mediated gene silencing may occur in part through sequestration of compacted chromatin in phase-separated HP1 droplets[13]. Studies have also shown that HP1 is a multi-functional protein that can act as a key regulator of some euchromatic genes that regulate tamoxifen resistance and epigenetic modifications in breast cancer cells[14]. However, there is limited knowledge about the roles of HP1 proteins in cancer drug resistance, especially in hematological malignancies including MM.

In the current study, we use the Stable Isotope Labeling with Amino Acids in Cell culture (SLIAC) assay to screen differentially expressed proteins between proteomics and acetylomics in our previously established BTZ-resistant (BR) MM cells, and identify that high levels of HP1γ are accompanied by low acetylation levels in the BR-MM cells. We investigate the underlying roles of HP1γ in regulating DNA repair through the formation of LLPS, and evaluate the efficacy of targeting HP1γ acetylation in overcoming resistance to PIs through in vivo and in vitro experiments.

## Results

### SILAC-labeled proteomics and acetylomics assay identify changes in HP1γ as a key feature of BR MM cells

To investigate the proteomics features of bortezomib-resistant (BR) MM cells that have been successfully established in our previous studies[15,16], we labeled the wild-type (WT) MM.1 S cells with normal L-lysine and BR MM.1 S cells with $^{13}C_6$-$^{15}N_2$-L-Lysine for six passages and then for the SILAC assay (Fig. 1a). BR MM.1 S cells yielded 176 differentially expressed proteins (DEP) and 167 differentially acetylated proteins, of which 33 proteins overlapped (Fig. 1b). Of these proteins, HP1γ was one of the top changed ones (Fig. 1c and Supplementary Fig. 1a). Interestingly, gene ontology (GO) analyses of these DEPs showed enhanced binding of histone deacetylase and response to drug (Supplementary Fig. 1b). Moreover, analysis of the acetylomic sites showed that the acetylation ratio of lysine 5 (HP1γ-K5, MASNK(Ac)TTL) was one of the highest of the acetylated lysine residues (Fig. 1d, e and Supplementary Fig. 1c). Consistent with these results, western blotting and the immunoprecipitation (IP) assay confirmed the upregulation of total protein and suppression of HP1γ acetylation in the BR MM cells (Fig. 1f, g). These results indicate that changes in HP1γ acetylation may be a key feature of BR MM cells.

### HP1γ is associated with treatment response in the clinic and sensitivity to BTZ in vitro

Clinically, when analyzing the coMMpass database from the Multiple Myeloma Research Foundation (MMRF), we found that high HP1γ expression was correlated with poor overall survival (OS), as the median OS of HP1γ-high and HP1γ-low patients was 70 and 91 months, respectively (Fig. 1h). Moreover, HP1γ expression in patients who had a complete response (CR) to PI-based regimens was significantly lower than that in patients with partial response (PR) (Fig. 1i). Similarly, analysis of another cohort of 542 MM patients (GSE9782) also showed that HP1γ expression was significantly lower in responders than non-

responder (Fig. 1j). We examined the expression of all three members of the HP1 family, HP1α (CBX5), HP1β (CBX1), and HP1γ (CBX3) in the BR MM.1 S cells and found that only HP1γ was significantly elevated (Supplementary Fig. 1d). Clinically, we found that HP1γ protein level was clearly suppressed after a PI-based regimen in two patients with CR, but was markedly increased in two patients with disease progression (Fig. 1k and Supplementary Fig. 1e). As expected, when the expression levels of HP1γ in MM cells were successfully suppressed or augmented via shRNA or overexpression (Supplementary Fig. 2a, b), the $IC_{50}$ values for BTZ were significantly suppressed or increased, respectively (Fig. 1l and Supplementary Fig. 2c). Additionally, manipulation of HP1γ expression also altered the PIs-induced apoptosis of MM cells, as shown by the flow cytometry assay and cleaved PARP by Western blotting (Fig. 1m and Supplementary Fig. 2d, e). Notably, HP1γ overexpression promoted the proliferation of MM cells (Supplementary Fig. 2f). Collectively, these data suggest that HP1γ is closely correlated with the clinical outcomes of MM patients and plays a pivotal role in regulating the sensitivity of MM cells to PIs.

### HDAC1 directly interacts and deacetylates HP1γ protein at lysine 5 to enhance protein stability during the development of MM resistance

To determine the acetylation modification mechanism of HP1γ, we ectopically overexpressed flag-tagged HP1γ in HEK293T cells and immunoprecipitated the HP1γ complex for mass spectrometry (MS). We found that histone deacetylase 1 (HDAC1) and HDAC2 were all present in the components of the HP1γ complex, that are well-known protein deacetylation enzymes[17] (Fig. 2a). The interaction between endogenous HP1γ and HDAC1 or HDAC2 could also be detected in MM cells (Fig. 2b), but the interaction between HDAC1 and HP1γ was more potent than that between HDAC2 and HP1γ; thus, we immunoprecipitated HDAC1-flag fusion protein from LP-1 cells, and confirmed their interaction (Fig. 2c). We constructed and purified exogenous GST-fusion HP1γ proteins and HDAC1-flag fusion protein to confirm their direct interaction in vitro (Fig. 2d, e). After overexpressing HDAC1 in a gradient, a dose-dependent increase in HP1γ level was observed (Fig. 2f). Furthermore, we found elevated expression of HDAC1, but not HDAC2 in the BR MM cells (Fig. 2g). Silver staining and IP showed that the interaction between HP1γ and HDAC1 was enhanced in the BR MM cells (Fig. 2h, i).

To further decipher whether HP1γ protein stability is regulated by acetylation in BR MM cells, we treated WT and BR MM cells with cycloheximide (CHX) for up to 8 h to cease protein synthesis, and found that degradation of HP1γ in BR cells was markedly delayed compared to WT cells (Fig. 2j), which phenotype was similar to what was observed in MM cells stably expressing ectopic HDAC1 (Supplementary Fig. 3a). We overexpressed HDAC1 and its vector control in the HP1γ-expressing HEK293T cells, and confirmed the deacetylation of HP1γ protein at lysine 5 by MS (Supplementary Fig. 3b, c). Therefore, we further mimicked K5 deacetylation by mutating lysine to arginine (K5R) and mimicked K5 acetylation by mutating lysine to glutamine (K5Q), and found that K5R-HP1γ mutants became more stable, whereas K5Q-HP1γ was more weakened in HEK293T cells (Supplementary Fig. 3d). Notably, K5R mutants effectively enhanced the stability of HP1γ protein (Fig. 2k) in MM cells, and administration of romidepsin (Rom) depleted HP1γ stability in BR MM cells (Supplementary Fig. 3e). Compared with LBH589, the first generation pan-HDAC inhibitor, Rom has a more obvious degradative effect on HP1γ protein (Supplementary Fig. 3f).

To clarify which degradation pathway is involved in regulating HP1γ degradation, we administered the proteasome inhibitor MG132 and lysosomal inhibitor LLOME in MM cells, and validated that HP1γ degradation was proteasome-dependent (Supplementary Fig. 3g). We next determined whether deacetylation of HP1γ could affect HP1γ protein ubiquitination from degradation. Our results showed that

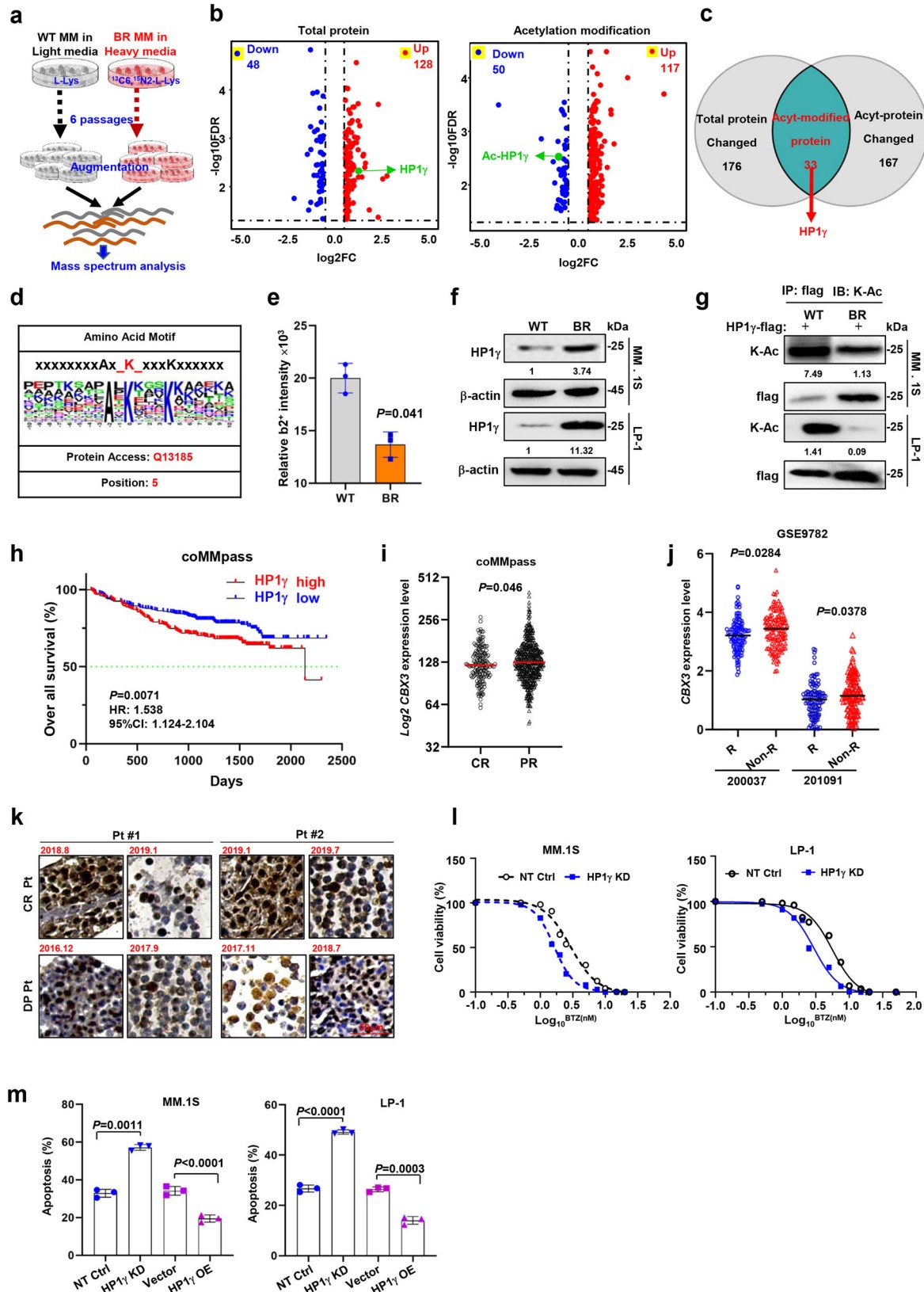

HDAC1 overexpression attenuated, but administration of Rom markedly enhanced HP1γ ubiquitination (Supplementary Fig. 3h, i). Similarly, K5R diminished, but K5Q accelerated the ubiquitination of HP1γ protein, indicating that K5 acetylation is critical to HP1γ ubiquitination (Fig. 2l). We further deciphered all possible linking patterns of

polyubiquitination on HP1γ protein, and found that K11-linked ubiquitin chain played a dominant role in HP1γ polyubiquitination modification (Fig. 2m, n). Collectively, these results suggest that deacetylation of HP1γ by HDAC1 at K5 hinders the K11-linked polyubiquitination of HP1γ and its instability.

**Fig. 1 | HP1γ and its acetylation modification are associated with BTZ resistance in MM cells. a** Schematic illustration of the manipulation of SILAC assay in the BR and WT MM.1 S cells. **b** Shown were 176 differential expressed proteins including 48 downregulated and 128 upregulated proteins (cutoff, 1.5 folds), furthermore, 167 proteins whose acetylation level were changed involving 50 downregulation and 117 upregulation (cutoff, 1.5 folds) in the BR MM.1 S cells compared to the WT cells detected by SILAC assay. **c** Venn diagram showed the most prominent change of protein-HP1γ in proteomics and acetylated omics of WT and BR MM cells. **d** Analysis of acetylated peptides identified specific acetylation motifs associated with lysine acetylation. **e** Relative b2+ intensity for K5 acetylation in WT and BR MM cells by mass spectrum (mean ± s.d.; $n = 1$). **f** Western blots showed the protein and acetylation levels of HP1γ in BR and WT MM cells. **g** Co-immunoprecipitation (Co-IP) assay showed the protein and acetylation levels of HP1γ in BR and WT MM cells with HP1γ-flag stably overexpression. Input, 2% lysate. IP, M2-flag antibody.
**h** Correlation of *CBX3* (HP1γ) expression with overall survival (OS) in 586 patients after receiving BTZ-based treatment regimens from the coMMpass cohort.
**i** Expression of *CBX3* in patients acquired with complete response (CR) ($n = 114$) and partial response (PR) ($n = 125$) in coMMpass cohort, and (**j**) in MM patients with responses (R, $n = 84$) and without response (non-R, $n = 125$) to BTZ-based regimens in the cohort GSE2658. 200037 and 201097, two probes in gene array.
**k** Representative immunohistochemical staining of HP1γ protein in specimen from two patients acquired complete response (CR) or disease progression (DP) after treatment. **l** Alteration of IC50 to BTZ treatment in the NT Ctrl and HP1γ KD cells ($n = 3$). **m** Quantification of the percentage of apoptotic cells in the Vector with HP1γ OE cells and NT Ctrl with HP1γ KD cells induced by 10 nM BTZ for 24 h by flow cytometry assay and analyzes the difference between two groups for $n = 4$ independent experiments (mean ± s.d.). *P* values were determined by Pearson's coefficient and log-rank test (**h**) and Student's *t* test (**i, j, m**). Source data are provided as a Source data file.

## HP1γ enhances MM drug resistance through DNA repair

Given that HP1γ plays a critical role in regulating sensitivity to PIs, we next investigated how HP1γ regulates MM drug resistance. GO analysis showed that the DNA repair pathway was significantly enriched in the BR MM cells. Indeed, we found that BR MM cells were more tolerant to BTZ-induced double-strand DNA breaks (DSBs) and apoptosis, as evidenced by obviously less phosphorylated H2AX (γH2AX), comet tail and cleaved PARP-1 (Fig. 3a and Supplementary Fig. 4a). In addition, using DNA topoisomerase I inhibitor Camptothecin (CPT) induced DSBs model, we also confirmed that BR MM cells indeed exhibited more potent DNA repair ability, as shown by the decreased γH2AX (Fig. 3b). On the contrary, when HP1γ was depleted in the BR MM cells, CPT could induce remarkable γH2AX accumulation and obvious comet tails (Fig. 3c, d and Supplementary Fig. 4b). Because homologous recombination repair (HR) and nonhomologous end joining (NHEJ) are both involved in DSB repair[18], we determined by which type HP1γ is involved in DNA repair using a flow cytometry-based reporter[19], and found that HP1γ overexpression is principally mediated by HR to repair the DSBs (Fig. 3e). Actually, main members enrolled in HR repair pathways, such as p-ATM and p-P95/NBS1 could be remarkably induced by CPT in the BR MM cell compared with in WT MM cells (Supplementary Fig. 4c), which phenotype was similar with overexpressing of HP1γ in MM cells (Supplementary Fig. 4d). At the same time, BR and ectopic HP1γ overexpression MM cells exhibited active S and G2 phases of the cell cycle, which could provide a template for HR (Supplementary Fig. 4e, f). Moreover, administration of the p-ATM inhibitor, YU238259, elicited remarkable cell apoptosis when combined with BTZ (Supplementary Fig. 4g), or in HP1γ depletion MM cells (Supplementary Fig. 4h). Collectively, these data suggest that the repair of DSBs is mediated by HR in BR MM cells.

Mechanistically, our mass spectrum assay data in Fig. 2A showed that TOPBP1, MDC1 and TRIM28 are partners of the HP1γ complex, that have been known to promote DNA repair[20–22]. We detected a tight interaction between HP1γ-flag and MDC1-GFP protein, but weak interaction with TOPBP1 (Fig. 3f and Supplementary Fig. 4i). We examined MDC1 and TRIM28 expression in WT and BR MM cells, and found that only MDC1 expression was changed (Supplementary Fig. 4j). In addition, the recruitment of endogenous HP1γ induced by UVA laser microdissection was severely impaired in MDC1 knockdown MM cells (Fig. 3g), and in HEK293T cells (Supplementary Fig. 4k, l). Taken together, these data suggest that MDC1 binds HP1γ to exert the effect of DNA repair in order to enhance the BTZ resistance of MM cells.

## Deacetylation at K5 enhances nuclear condensation of HP1γ

When endogenous HP1γ proteins were detected by immunofluorescence, we found that they were shown as separated puncta rather than in a diffused status in MM cells, and this phenomenon was more pronounced in BR MM cells, suggesting a possible property of a protein when entering the liquid-liquid phase separation (LLPS) state (Fig. 4a). Moreover, such puncta were also observed in live HEK293T-cells expressing a GFP-HP1γ fusion protein, and when it was treated with 1,6-hexanediol (1,6-Hex), the formation of GFP-HP1γ puncta was dispersed (Supplementary Fig. 5a); when bleached by a 488-nm laser, the puncta were re-aggregated quickly (Supplementary Fig. 5b). To clarify whether deacetylation modification has any influence on the nuclear condensation of HP1γ, we co-expressed HP1γ-GFP with or without HDAC1-flag in HEK293T cells (Supplementary Fig. 5c), and found kinetic recovery of GFP-HP1γ fluorescence after bleaching recovered more rapidly in the HDAC1 overexpressing cells than in the vector control (Fig. 4b). Akin to the exogenously expressed HP1γ, the endogenously HP1γ puncta were more abundant in HDAC1-flag overexpressing HEK293T cells than in the vector control (Supplementary Fig. 5d). Moreover, in MM cells, when the HDAC1 was forcedly expressed, the nuclear condensation of HP1γ was remarkably promoted (Fig. 4c).

We predicted the HP1γ protein structure and found two putative intrinsically disordered regions (IDRs), where the above probed K5 and other K10, K13, K14, K18, K20, K21, and K34 acetylation sites were predominantly localized (Supplementary Fig. 5e). To test the importance of the deacetylated IDR1 region for HP1γ phase separation, we constructed GFP-fused IDR1 with lysine 5 site mutating to arginine (K5R) or glutamine (K5Q), or mutating all lysine to glutamines (KallQ) to measure droplet formation in vitro (Supplementary Fig. 5f–h). Intriguingly, K5R mutation significantly augmented the droplet formation, however K5Q remarkably attenuated and KallQ almost extirpated droplet formation of GFP-HP1γ-IDR1 fusion protein (Fig. 4d). The turbidity assay also showed that K5Q mutant had lower turbidity than the GFP-HP1γ-IDR1-WT fusion protein, but K5R strengthened the turbidity of GFP-HP1γ-IDR1 fusion protein (Fig. 4e). Strikingly, either the GFP-HP1γ-full length (FL)-WT, FL-K5R and FL-K5Q proteins expressed in HEK293T in vivo, or purified GFP-HP1γ-IDR1-WT, IDR1-K5R and IDR1-K5Q fusion proteins in vitro, kinetic recovery of the puncta and droplets after bleaching showed that the majority of foci fluorescence recovered much quickly for the K5R mutant (Fig. 4f, g and Supplementary Fig. 6a–c). In an in vitro deacetylation reaction system, when purified HDAC1 protein was co-presented with HP1γ protein, the droplet formation of HP1γ was remarkably enhanced (Fig. 4h), and presence of HDAC1 yielded a significantly stronger turbidity than mock control (Fig. 4i). These results show that deacetylation at K5 is sufficient to robustly alter the condensation formation of HP1γ.

We further evaluated the importance of the nuclear condensation of HP1γ in DNA repair and MM drug resistance. We found overexpression of HP1γ-K5R, but not HP1γ-K5Q, could dramatically improve the ability of HR repair compared with the HP1γ-WT (Fig. 4j and Supplementary Fig. 6d). Surprisingly, when we immunoprecipitated the HP1γ-WT, -K5R and -K5Q proteins, we did not observe distinct changes in their interactions with MDC1, HDAC1 and H3K9me3

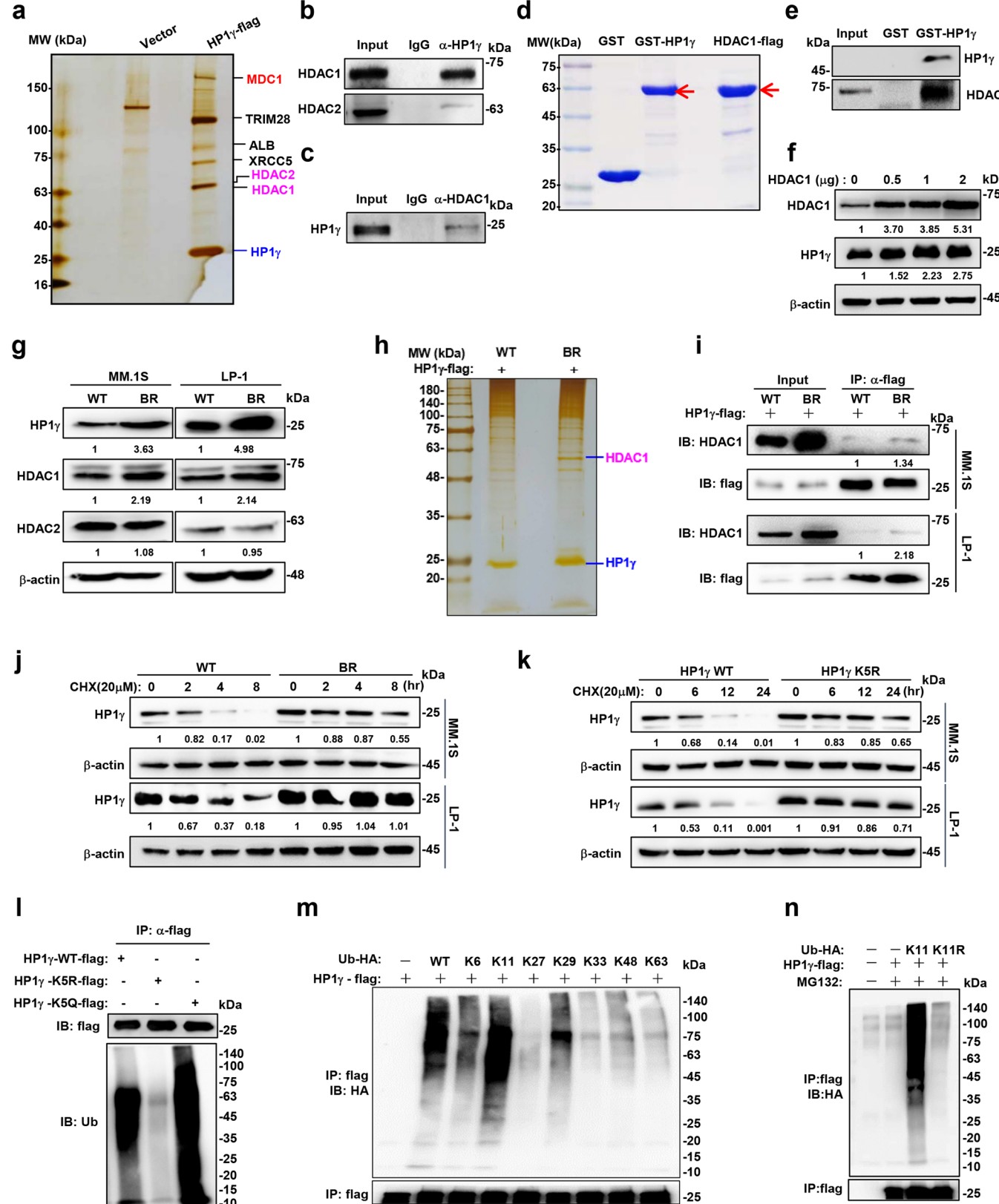

(Supplementary Fig. 6e), despite that overexpression of HP1γ-K5R in MM cells indeed elicited stronger chemoresistance and proliferation (Supplementary Fig. 6f, g). Taken together, these results suggest that the nuclear condensation of HP1γ plays a critical role in drug resistance through DNA repair in MM cells, and this role does not depend on changes in the protein-protein interaction.

## Interaction with MDC1 promotes HP1γ phase separation

Our previous study demonstrated that interaction with partner proteins enhances the phase separation property of the master protein[16]. To clarify the relationship between interaction with MDC1 and nuclear condensation of HP1γ, we firstly evaluated the possibility of LLPS in MDC1 protein, and found that this protein has at least three putative

**Fig. 2 | HDAC1 deacetylates and stabilizes HP1γ protein. a** Silver staining for flag-pulldown of flag-HP1γ complex in HEK293T cells. **b** Co-IP assay to show interactions between endogenous HP1γ with HDAC1 and (**c**) endogenous HDAC1 with HP1γ in MM.1 S cells. Input, 2% whole cell lysate. **d** Coomassie blue staining to show in vitro expression of GST-fusion HP1γ protein and in vivo expression of flag-HDAC1 protein in HEK293T. **e** GST-pulldown assays to show the interaction between purified HDAC1-flag and GST-tagged-HP1γ protein in vitro. Input, purified HDAC1-flag. **f** Levels of HP1γ protein in HEK293T cells with increasing overexpression of HDAC1 by western blotting. **g** Western blotting shows the alteration of HP1γ, HDAC1 and HDAC2 levels in WT and BR MM.1 S and LP-1 cells. **h** Silver staining for flag-pulldown

in the WT and BR LP-1 cells infected with lentivirus carrying HP1γ-flag for 72 h. **i** Alteration of interaction between HP1 and HDAC1 in WT and BR LP-1 and MM.1 S cells infected with HP1γ-flag for 72 h. Input, 2% lysate. **j** Degradation rate of HP1γ protein in WT and BR MM.1 S and LP-1 cells treated with 20 μM cycloheximide (CHX). **k** Western blotting shows the HP1γ protein level in the HEK293T cells transfected with flag-tagged HP1γ WT, K5R and K5Q. **l** Ubiquitination status of HP1γ-WT-flag, HP1γ-K5R-flag or HP1γ-K5Q-flag in HEK293T cells. **m** Polyubiquitination linkage features of HP1γ-flag mapping by positive mutations of ubiquitin in HEK293T cells. **n** K11-linked polyubiquitination of HP1γ-flag in HEK293T cells. Source data are provided as a Source data file.

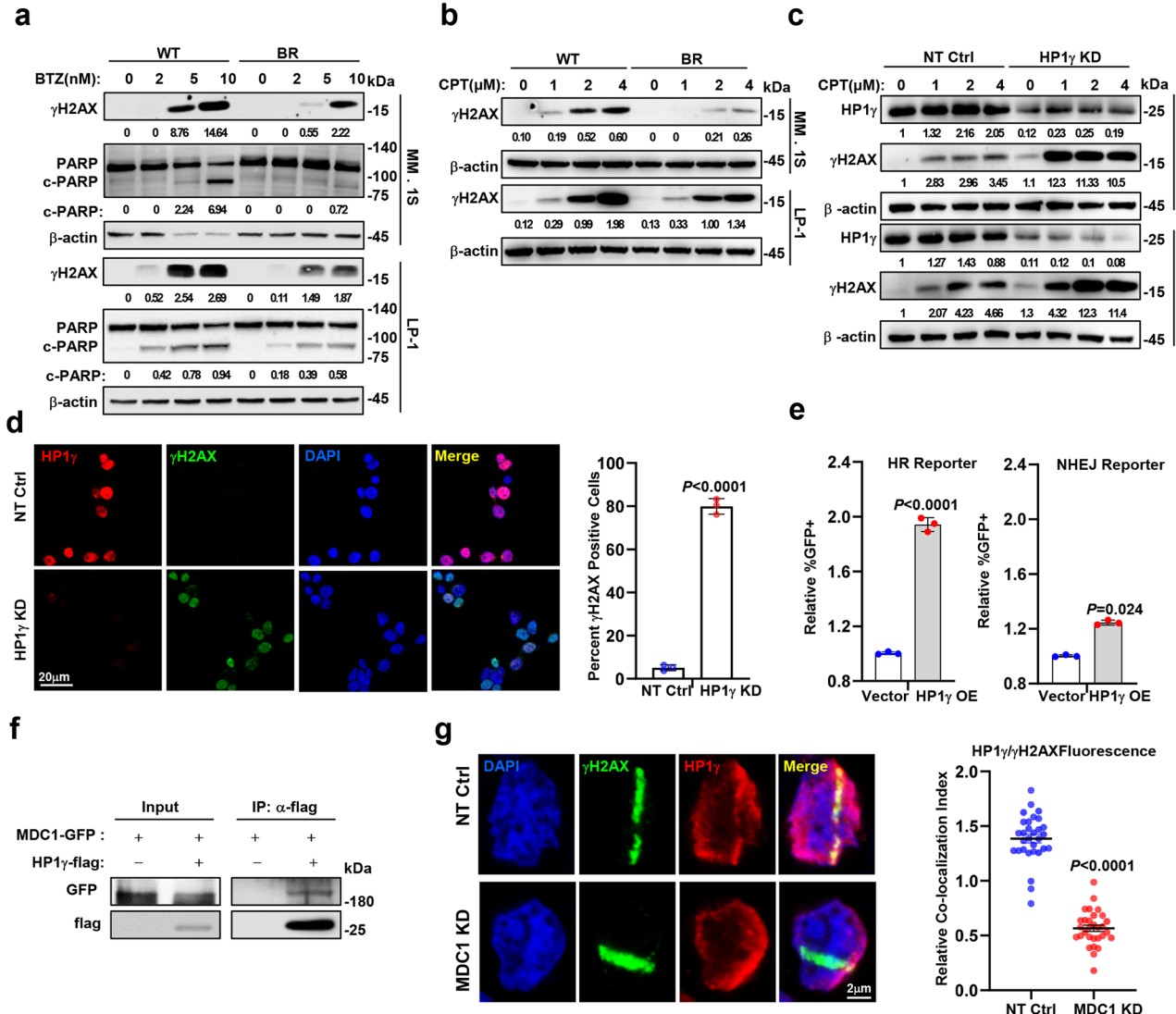

**Fig. 3 | HP1γ facilitates homologous recombination of DNA repair. a** Western blotting shows γH2AX and cleaved PARP in WT and BR MM cell treated with BTZ for 24 h. **b** Levels of γH2AX in WT and BR MM cells treated with camptothecin (CPT, 0−4 μM) for 24 h. **c** Western blotting shows γH2AX in MM cells with HP1γ KD treated with gradient concentration CPT for 4 h. **d** Left: immunofluorescence assay shows the level of γH2AX in WT and BR MM cells treated with CPT (2 μM) for 4 h. Right: quantification of γH2AX foci. At least 100 cells were counted, and n = 3 independent experiments. Data represent mean ± S.E.M., student's t test. **e** Quantitative assessment of HR and NHEJ activities in HEK293T cells expressing vector control or HP1γ

expressing plasmid via flow cytometry assay for the percentage of GFP+ cells among RFP+ cells. Quantitative data are shown as means ± s.e.m., student's t test, and n = 3 (biologically independent experiments). **f** Co-IP assay shows bilateral interactions between endogenous HP1γ and MDC1. Input, 2% whole cell lysate. **g** Left: immunofluorescence assay demonstrating the HP1γ recruited to DNA damage regions in MDC1-KD CAG cells. Right: the graph represents the mean intensity of the HP1γ protein on the γH2AX track. Error bars represent S.E.M., n = 3 (biologically independent experiments), and differences relative to NT-Ctrl were calculated using student's t test. Source data are provided as a Source data file.

IDR regions (Supplementary Fig. 7a). When ectopic GFP-tagged MDC1 was expressed in HEK293T cells, the discrete puncta effectively completed fluorescence recovery after photobleaching (FRAP) (Fig. 5a). When the cells were treated with 1,6-hexanediol, formation of

exogenously expressed GFP-MDC1 puncta was suppressed and fluorescence of the GFP-MDC1 was diffused (Fig. 5b), which highlight the LLPS-like character of MDC1. Because protein-protein interactions through IDR regions enhance their capacity to form the LLPS[23], we

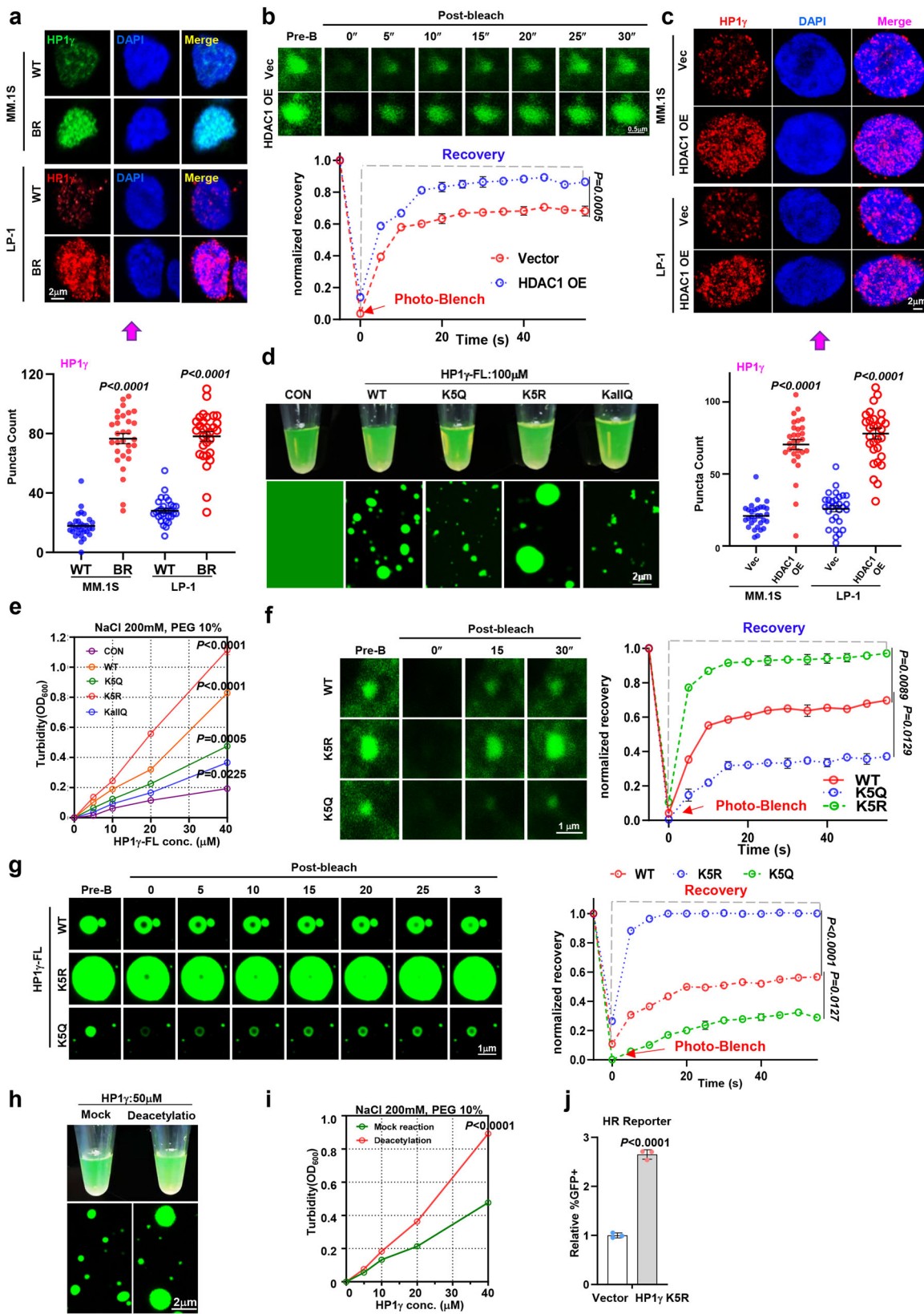

designed GFP-tagged truncations of the IDRs of MDC1 for GFP-pulldown (Fig. 5c). When these GFP-tagged truncations were co-immunoprecipitated with GST-HP1γ fusion protein, only the N1, N2 and N3 truncations, where IDR1 and IDR2 were localized, interacted with HP1γ protein (Fig. 5d and Supplementary Fig. 7b). We also constructed a series of HP1γ truncations for flag pulldown, and confirmed that the

CSD domain of HP1γ was responsible for interaction with MDC1 (Supplementary Fig. 7c, d).

Since the above Co-IP data suggest that IDR1 and IDR2 of MDC1 are dominant regions that interact with HP1γ, we constructed mCherry and GST double tagged IDR1 (GST-MDC1-mC-IDR1, 143-339aa) and IDR2 (GST-MDC1-mC-IDR2, 354-539aa) (Supplementary Fig. 7e). As

**Fig. 4 | Acetylation impairs nuclear condensation of HP1γ in vivo *and vitro*.**
**a** Upper: representative images and quantification of puncta count for endogenous foci of HP1γ in WT and BR MM.1 S and LP-1 cells. Error bars represent S.E.M., *n* = 3 (biologically independent experiments). Lower: quantification of puncta count is shown in images. **b** FRAP assay (upper) and kinetic recovery times (lower) for a HP1γ-GFP focus by 488 nm laser for 5″bleaching and 40″recovery in HEK293T cells transfected with HDAC1 or vector control (mean ± s.d.; *n* = 3 independent experiments). **c** Upper: Representative images for HP1γ foci in Vector and HDAC1 overexpression MM cells (*n* = 3). Lower: Quantification of puncta count is shown in images. **d** Visualization of turbidity associated with droplet formation in vitro for the GFP-vector control (Ctrl), (HP1γ-IDR1)-WT, -K5Q, K5R or -KallQ. **e** Turbidity (OD600) of WT, K5Q, K5R, and KallQ mutants of HP1γ-IDR1 in 200 mM NaCl and 10% PEG was measured (mean ± s.d.; *n* = 3 independent experiments). **f** FRAP assay for the recovery of fluorescence intensity of WT-, K5R- and K5Q-HP1γ-FL in HEK293T cells (mean ± s.d.; *n* = 3 independent experiments). **g** Microscopy images of GFP fusion proteins of WT-, K5Q- and K5R-HP1γ-FL before and after partial droplet photobleaching in vitro (mean ± s.d.; *n* = 3 independent experiments). **h** Visualization of turbidity associated with droplet formation in vitro for mock reaction and deacetylated HP1γ protein. **i** Turbidity (OD600) of mock reaction and deacetylated HP1γ protein in 200 mM NaCl and 10% PEG was measured (mean ± s.d.; *n* = 3 independent experiments). **j** Quantitative assessment of HR activity in the HEK293T cells expressing vector control or K5R-HP1γ via flow cytometry assay for the percentage of GFP+ cells among RFP+ cells (mean ± s.d.; *n* = 3 independent experiments). *P* values were determined by one-way ANOVA (**b, i**), two-way test (**e, f, g**) and Student's t test (**a, c, j**). Source data are provided as a Source data file.

detected by GST-pulldown, both MDC1-IDR1 and MDC1-IDR2 tightly interacted with HP1γ (Fig. 5e). Droplet formation assays revealed that only MDC1-IDR2 could form phase-separated droplets, but IDR1 and IDR3 showed very limited droplet formation capacity (Fig. 5f and Supplementary Fig. 7f, g); dynamically, the liquid droplets of MDC1-IDR2 at high protein concentration could recover completely after FRAP (Fig. 5g). These data confirm the LLPS property of MDC1. To test whether MDC1-IDRs incorporated into HP1γ phase-separated condensates, we created an in vitro MDC1-mC-IDRs and HP1γ-GFP co-interaction droplet system. Interestingly, we observed that mixing with HP1γ-GFP remarkably increased the droplet formation of MDC1-IDR2, and even promoted the droplet formation of MDC1-IDR1, which was unable to form droplets solely (Fig. 5h). Turbidity assay of the fusion proteins indicated that the MDC1-IDR1 without LLPS formation capacity decreased the turbidity of the HP1γ-GFP. By contrast, MDC1-IDR2 and HP1γ all have LLPS formation capacity could enhance each other's turbidity mutually (Fig. 5i). Immunofluorescence staining also showed that MDC1-GFP was co-localized with HP1γ-mCherry on the discrete puncta (Fig. 5j). In the BR-MM cells, more HP1γ puncta were in correspondence to more MDC1 puncta (Supplementary Fig. 7h), inhibition of HP1γ deacetylation induced less HP1γ condensation (Supplementary Fig. 7i), and treatment of CPT elicited more and larger HP1γ and MDC1 puncta (Supplementary Fig. 7j). Collectively, these data suggest that both MDC1 and HP1γ possess the property of LLPS, and their interaction enhances the nuclear condensation of each other.

## HP1γ regulates chromatin accessibility of genes governing drug sensitivity

Because HP1γ is distributed in both euchromatin and heterochromatin and is associated with active transcriptional elongation[10], we investigated how HP1γ regulates key genes responsible for chemoresistance in MM cells. RNA sequencing (RNA-seq) analysis identified 1149 upregulated and 1281 downregulated genes in the HP1γ overexpressed MM cells (Fig. 6a and Supplementary Data 1). The Kyoto Encyclopedia of Genes and Genomes (KEGG) analysis indicated that upregulated genes were enriched for the drug resistance and replication and DNA repair, consistent with the role in enhancing the BTZ resistance in MM cells (Fig. 6b). To further interpret the function of HP1γ in regulating genes responsible for chemoresistance, we mapped the genome-wide distribution of HP1γ by ChIP-seq in the BR and HP1γ-OE MM cells compared with their parental controls respectively, and found a considerable distribution of HP1γ was increased on the gene promoter, especially the promoters less than 1 kb (Fig. 6c, d). Our ChIP-seq of HP1γ showed 17,246 and 10,023 genes in the BR and WT MM cells, respectively (Supplementary Fig. 8a), and analysis of the differently bound genes (DBG) in the BR cells showed replication and repair, transcription, signaling molecules and transduction, cancer drug resistance processes were enriched by KEGG analysis (Supplementary Fig. 8b). Integrated with RNA-seq data, 188 genes were intersected in the DBGs and HP1γ upregulated genes (Fig. 6e and Supplementary Data 2). Of these intersected genes, we evaluated 25 of them in the BR

MM cells compared with the WT cells, and identified *FOS, JUN* and *CD40* (Supplementary Fig. 8c), that have been reported to regulate-drug resistance and DNA repair in MM cells and other cancers[24–27]. Firstly, we excluded the direct effect of HDACi on these genes, since HDACi only affected expressions of *CD40, JUN* and *FOS* when endogenous HP1γ was in presence, but not in HP1γ depletion BR-MM cells (Supplementary Fig. 8d). We observed that expressions of these genes were all significantly upregulated in the HP1γ overexpressing MM cells and BR-MM cells (Fig. 6f, g), and importantly, these genes were also upregulated in 6 cases of RRMM patients after BTZ-based treatment, together with elevated HP1γ (Fig. 6h). Moreover, metagene analysis and representative gene tracks showed that the normalized tag density of HP1γ was significantly increased around TSS of *FOS, CD40*, and *JUN* (Fig. 6i and Supplementary Fig. 9a, b). To further interpret how HP1γ regulates these genes' transcriptional expression, we detected the redistribution of H3K36me3 and H3K9me3 on promoters of these genes by ChIP-qPCR, and found that enrichment of H3K36me3 was enhanced, however H3K9me3 was attenuated significantly in HP1γ overexpressing MM cells and BR cells (Fig. 6j, k and Supplementary Fig. 9c–f). Functionally, knockdown of these genes significantly re-sensitized the BR-MM cells to BTZ treatment (Supplementary Fig. 9g), and forced expression of these genes partially rescued resistance to BTZ in the HP1γ-KD MM cells (Supplementary Fig. 9h). Clinically, high expressions of these genes all predicted significantly poorer PSF and OS in an independent MM cohort (Supplementary Fig. 9i, j).

Given that HP1γ remodels chromatin to facilitate gene transcription, we employed an assay for transposase-accessible chromatin with high-throughput sequencing (ATAC-seq) to investigate the impact of HP1γ on chromatin accessibility in MM cells. Global augmentation of chromatin accessibility in the HP1γ-OE cells was observed when compared to the vector control (Fig. 7a), and the chromatin remodeling occurred mainly at the key regions of promoters (≤1Kb), but was not significant in the distal promoters, gene body, nor untranslated regions (UTRs) (Fig. 7b); the similar alteration was also observed in the BR MM cells compared with WT cells, but the extent was not comparable (Fig. 7c, d). Specifically, the intensity of ATAC-seq signal at the TSS regions of *FOS, JUN* and *CD40* genes was augmented after HP1γ overexpression (Fig. 7e), which has also been observed in the BR-MM cells (Fig. 7f). Importantly, we analyzed the chromatin accessibility of primary CD138+ plasma cells from 3 relapsed MM patients after BTZ-based treatment, and verified that intensity of ATAC-seq signal at the TSS regions of these genes were all remarkably amplified (Fig. 7g). Together, these findings revealed that HP1γ positively regulated the chromatin accessibility of genes governing drug resistance in MM cells.

## Targeting HP1γ stability using HDAC inhibitor re-sensitizes BR MM cells to PI treatment in vivo and in vitro

To evaluate the effects of targeting HP1γ deacetylation on overcoming MM drug resistance, we examined the effects of an HDAC1/2 inhibitor, Rom, in vitro and in vivo. The combination of Rom significantly

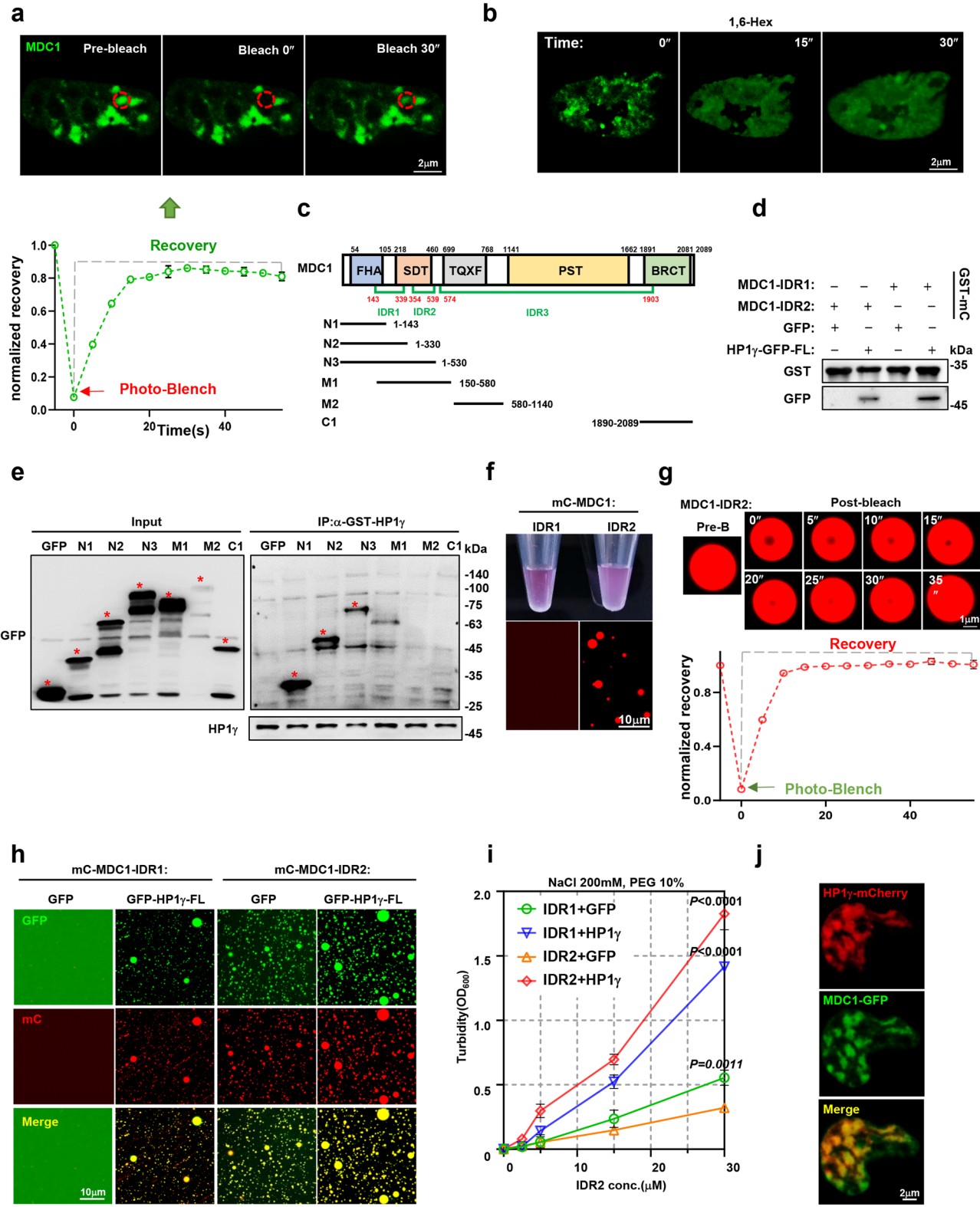

improved the sensitivity to BTZ treatment in BR MM cells, as evidenced by the enhanced apoptosis rate and the cleavage of PARP-1 in the Rom and BTZ combination group compared to the BTZ or Rom solo administration controls (Fig. 8a, b). Similarly, the combination of Rom and BTZ has a more cytotoxic effect on BR-MM cells than the combination of LBH598 and BTZ (Fig. 8c). Using an HP1γ expression stably silenced (HP1γ-KD) BR MM.1 S cells, we established a xenograft model in the NOD.Cg-Prkdc[scid]Il2rg[tm1Wjl]/SzJ (NSG) mice, and treated the mice with BTZ (0.5 mg/kg, every 3 d, subsequent BTZ doses in vivo

experiment as above). As expected, tumor growth was significantly suppressed only in the HP1γ KD derived xenografts but not in the parental control cells (Fig. 8d), and the overall survival of mice bearing HP1γ-KD BR MM.1 S cells improved significantly compared to the non-target control (NT Ctrl) (P < 0.0001) (Fig. 8e). Similar to the effect of HP1γ interference, combination administration of Rom (1 mg/kg, every 2 d, subsequent BTZ doses in vivo experiment as above) and BTZ also markedly suppressed BR-MM.1S-derived tumor growth (Fig. 8f), and significantly improved survival (P = 0.0002) (Fig. 8g). Meanwhile,

**Fig. 5 | Interaction with MDC1-IDR enhances nuclear condensation of HP1γ.**
**a** FRAP assay for a MDC1-GFP focus (red circle) by 488 nm laser for 5″bleaching and 30″recovery in HEK293T cells (mean ± s.d.; *n* = 3 independent experiments). **b** Dispersion of GFP-MDC1 foci fluorescence in HEK293T cells before and after treatment with 3% 1,6-hexanediol (1,6-Hex) at different time. **c** Mapping MDC1 truncations including MDC1-N1, -N2, -N3, -M1, -M2, -C1 fragments and MDC-IDR1, -IDR2 and -IDR3. **d** GST-pulldown assay shows the direct interaction between purified HP1γ-GFP and GST-tagged-mCherry-MDC1-IDR1 or -IDR2 protein in vitro. **e** GST pull-down assay shows interactions between GST-HP1γ and truncations of MDC1. Input, 2% lysate. *, exact positions of different GFP-MDC1 truncation fusion proteins. **f** Visualization of turbidity associated with droplet formation in vitro for 50 μM MDC1-IDR1 and MDC1-IDR2 solutions (top) and liquid droplets visualized by DIC microscopy. **g** Microscopy images of mCherry fusion proteins of MDC1-IDR2 droplets before and after partial droplet photobleaching (mean ± s.d.; *n* = 3 independent experiments). **h** Representative images of droplet formation of mCherry-MDC1-IDR1 or mCherry-MDC1-IDR2 in the presence or absence of GFP or HP1γ-GFP. **i** Turbidity (OD600) of MDC1-IDR2 in the presence of MDC1-IDR2, GFP or HP1γ-GFP in 200 mM NaCl and 10% PEG was measured (mean ± s.d.; *n* = 3 independent experiments). *P* values were determined by two-way ANOVA test. **j** Confocal live-cell fluorescence microscopy of HEK293T cells co-transfected with HP1γ-mCherry and MDC1-GFP for 48 h. Source data are provided as a Source data file.

tumor cells in the combination groups showed a higher rate of apoptosis as detected by the TUNEL assay (Fig. 8h), as well as higher acetylation status, ubiquitination level of HP1γ, suppressed HP1γ level and *FOS*, *JUN* and *CD40* expression (Fig. 8i–k). In addition, using an intra-bone growing MM model established in our laboratory[16], which can mimic the real bone microenvironment in MM patients, we treated the mice with BTZ solely or together with Rom after 2 weeks of inoculation, and found that the Rom-BTZ combination effectively alleviated bone lesion of mice destroyed by MM cells, as evidenced by a smaller size of trabecula separation at the metaphysis and diaphysis (Fig. 8l). Notably, using 6 RRMM patients' bone marrow unsorted bone marrow mononuclear cells, 3 samples were HP1γ-low and 3 samples were high in CD138+ plasma cells, we established the patient derived xenograft (PDX) model to assess the effects of the combination of BTZ and Rom. As expected, tumor burdens were extensively extenuated in the HP1γ-high ones, but those from HP1γ-low were not comparative (Fig. 8m). Taken together, these in vitro and in vivo phenotypes and biological data revealed that pharmacological targeting of HP1γ stability abrogates BTZ-induced drug resistance and stimulates bone-lesion recovery in mice.

## Discussion

In this study, we report a crucial role of post-translational modification of HP1γ in regulating of sensitivity to PIs in MM cells. We propose a mechanism that HDAC1-mediated deacetylation improves the nuclear condensation of HP1γ, consequentially reinforces the complex formation with MDC1 to favor DNA repair and altering the chromosomal accessibility of genes governing MM cell survival (Fig. 9). Translationally, our findings also suggest that targeting HP1γ stability using HDAC1 inhibitor resensitizes the anti-MM effect of PIs in resistant MM cells, which may benefit the management of relapse or refractory MM patients.

The current study proposes that HP1γ binding to MDC1 promotes drug resistance of MM through DNA repair. MDC1 contains the FHA domain and the C-terminal BRCA1 domain, both of them are reported to be involved in DNA repair response[28]. A previous study reported that recruitment of HP1γ to the DNA damage site requires the CSD but is independent of H3K9me3[29], however the detailed process has not been elucidated. Our study reveals that MDC1 plays a critical role in mediating the recognition of HP1γ into DNA damage sites. We propose that when DNA damage occurs, the MRE11-RAD50-NBS1 (MRN) complex sensitizes the damage and activates ATM, then H2AX is phosphorylated and binds to MDC1, and the HP1γ-Suv39H1-Kap-1 complex forms and binds to DNA damage sites to methylate H3K9, based on some known knowledge[30]. Thus, the findings of the current study provide a perspective for HP1γ to recognize non-H3K9me3-dependent DNA damage sites in MM cells.

Recently, ample researches have highlighted the importance of protein aggregations, such as liquid-liquid phase separation (LLPS) and nuclear condensation, in physiological and pathophysiological processes[31,32]. Our previous study also reported the LLPS of SRC-3 in mediating drug resistance to PIs in MM cell[16]. The acetylation and deacetylation of IDRs spatiotemporally regulate protein aggregation and impact membrane-less organelle formation in vivo[33]. However, protein aggregation patterns are diverse across the range of cells and treatments, the recognition and regulation are still obscure. In this study, we showed that deacetylation of HP1γ at the K5 site, which is the site localized at the first IDR of HP1γ, affected the nuclear condensation of HP1γ in BR MM cells. In addition to post-translational modification, when another LLPS-forming protein such as MDC1, interacts with HP1γ, it enhances the condensation of HP1γ. Specifically, we found that enhanced nuclear condensation of HP1γ exerted higher DNA repair capacity. Thus, targeting the key modifier such as HADC1, by manipulating gene expression or using small-molecular inhibitors, may suppress droplet formation and abrogate protein stability of HP1γ, and thus can overcome chemoresistance as a consequence. Collectively, the findings of this study provide at least two mechanistic insights into the roles of HP1γ in MM drug resistance, and also provide a perspective for better understanding how HP1γ bridges translational modifications with phase separation to promote chemoresistance in MM.

Translationally, we evaluated the effect of targeting HP1γ deacetylation using an HDAC1 inhibitor in overcoming BTZ-resistant MM. Our in vitro and in vivo data have provided strong evidence that HDAC1i has synergistic anti-MM effects with BTZ on BTZ-resistant MM cells, accompanied by accelerated HP1γ degradation, increased cell apoptosis, and better recovery of the bone lesion. Rom, as a single agent, exhibited evidence of M-protein stabilization, and several other patients experienced an improvement in bone pain and resolution of hypercalcemia[34]. HDACi has been reported to inhibit multiple processes of DNA repair, such as transcriptionally down-regulates a large number of DNA repair proteins, impairs the recruitment of DNA repair complex to the broken DNA sites, and affects a number of proteins for DNA repair through acetylation modification[35]. In support of this assertion, our results also suggest that administration of Rom suppresses the DNA repair process to overcome drug resistance via elevated acetylation level of HP1γ, as well as through interfering with the LLPS of HP1γ. Importantly, since several combinatorial HDACi and bortezomib trials on RRMM showed a modest effect on PFS and OS[36], our study also provides a clue to clarify the applicable population, who should have high HP1γ protein level in bone marrow biopsies after PI-based regimens, and thus contribute the development of precision medicine in MM management.

Overall, our present findings show the detailed assembly process of the HP1γ-MDC1 complex in recognizing the DSB sites and exerting DNA repair function, as well as how the nuclear condensation of HP1γ is regulated by deacetylation and facilitation by MDC1. The significance of our study is to provide a theoretical basis for developing treatment strategies targeting HP1γ to treat MM.

## Methods
### Ethic approvals
This study was approved by the Ethic Committee of Tianjin Medical University, and all the protocols were conformed to the Ethical

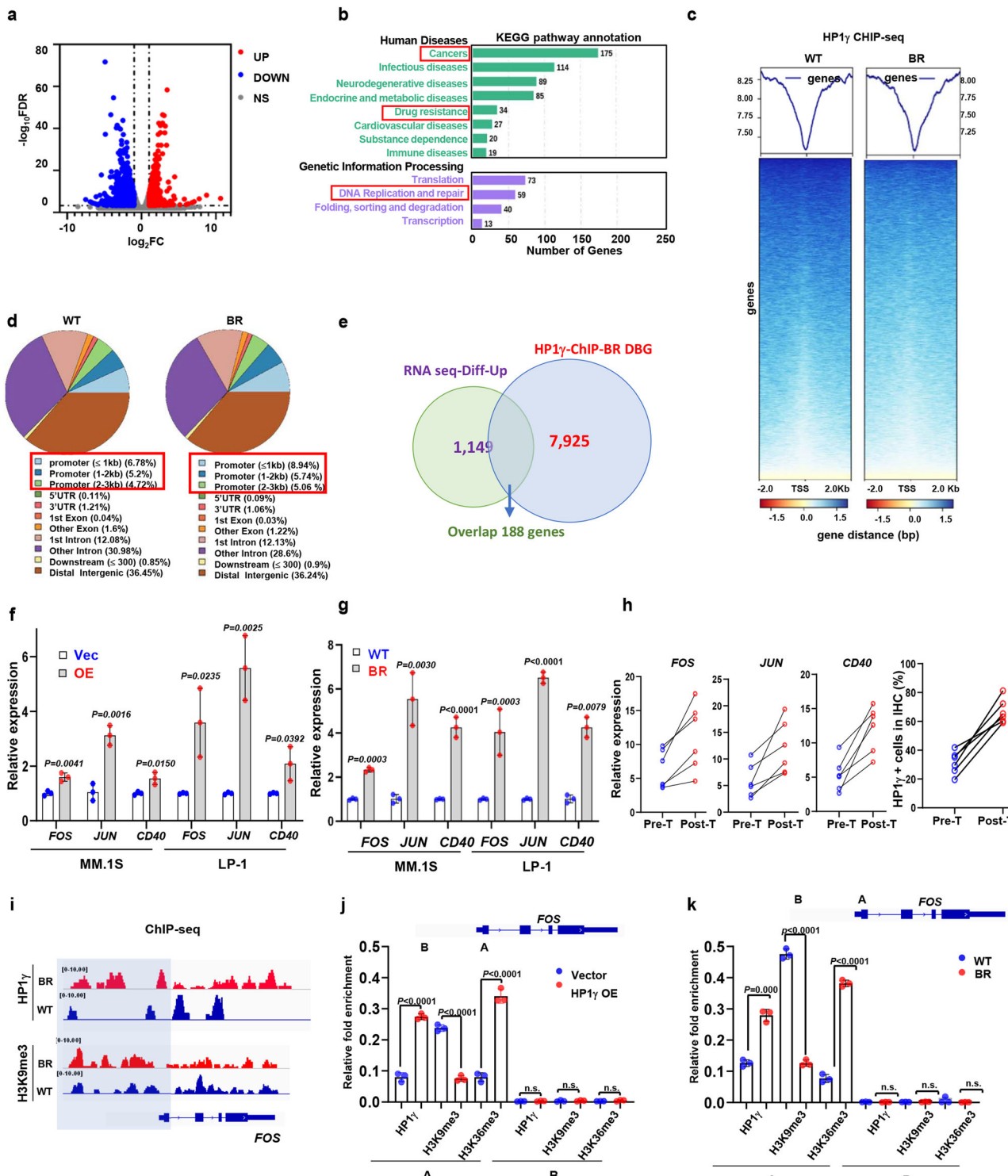

**Fig. 6 | HP1γ promotes expression of genes associated with MM drug resistance. a** Volcano plot of differentially expressed genes analyzed from bulk RNA-sequencing in HP1γ overexpressing LP-1 cells. Blue, downregulated genes; red, upregulated genes; gray, statistically non-significance genes. **b** KEGG analysis for differentially expressed genes with a $P < 0.05$ using DAVID methods. **c** ChIP-seq profile of genes enriched by HP1γ in WT and BR LP-1 cells ($n = 3$ biologically independent experiments). **d** Pie chart to show percentages of HP1γ binding on gene locations. **e** Venn diagram shows the number of overlapped genes between HP1γ-ChIP and HP1γ overexpression RNA seq. QPCR shows expression of *FOS, JUN* and *CD40* expressions in (**f**) the HP1γ overexpressing, (**g**) WT and BR MM.1 S and LP-1 cells, and (**h**) in 6 MM patients with disease progression before and after BTZ-based treatment (mean ± s.d.; $n = 3$ independent experiments). **i** Gene tracks showing representative ChIP-Seq profiles for the indicated proteins and histone marks at the *FOS* gene loci. ChIP-qPCR assay for the enrichment of HP1γ, H3K9me3 and H3K36me3 at gene loci of *FOS* in (**j**) LP-1 cells stably expressing vector or HP1γ, and (**k**) WT and BR LP-1 cells. Schematic representation of PCR primer design is provided. A, promoter region; B, intergenic region. Two-sided $P$ values were determined by Student's t test (**f, g, j, k**). Data indicate the mean ± SD, $n = 3$ independent experiments. Source data are provided as a Source data file.

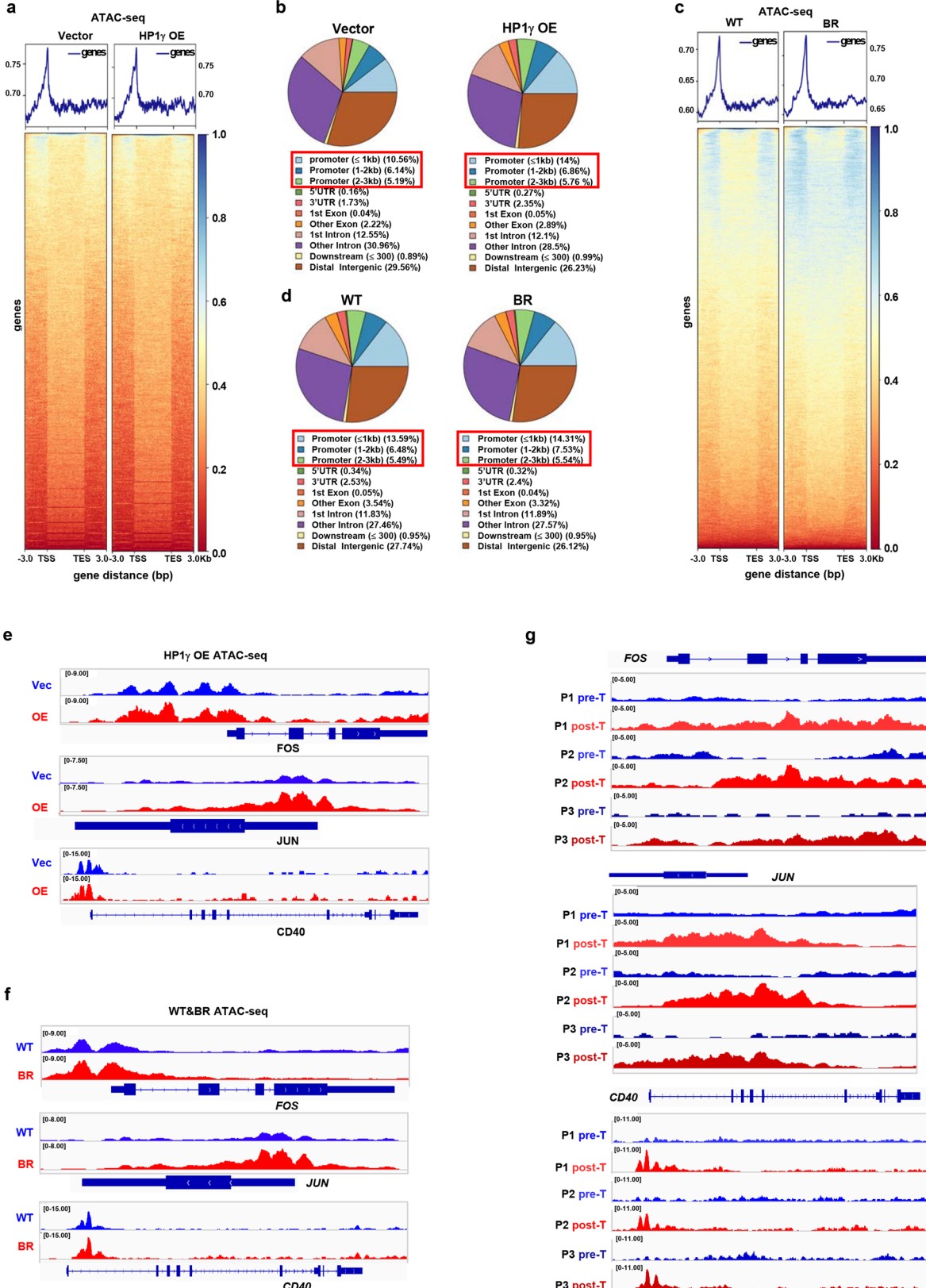

**Fig. 7 | HP1γ facilities chromatin accessibility of genes governing drug sensitivity. a** ATAC-seq profile of chromatin accessibility of genes in the Vector and HP1γ overexpressing LP-1 cells (*n* = 3 biologically independent experiments). **b** Genome-wide distribution of HP1γ accessibility regions in the Vector and HP1γ over-expressing LP-1 cells. **c** ATAC-seq profile of chromatin accessibility of genes in the WT and BR LP-1 cells (*n* = 3 biologically independent experiments). **d** Genome-wide distribution of HP1γ accessibility regions in the WT and BR LP-1 cells. **e** Gene tracks showing representative ATAC-Seq profiles at *FOS, JUN* and *CD40* gene loci in LP-1 cells stably expressing vector or HP1γ OE, and (**f**) WT and BR LP-1 cells. **g** Gene tracks shows representative ATAC-Seq profiles at *FOS, JUN* and *CD40* gene loci in 3 MM patients with disease progression before and after BTZ-based treatment. Source data are provided as a Source data file.

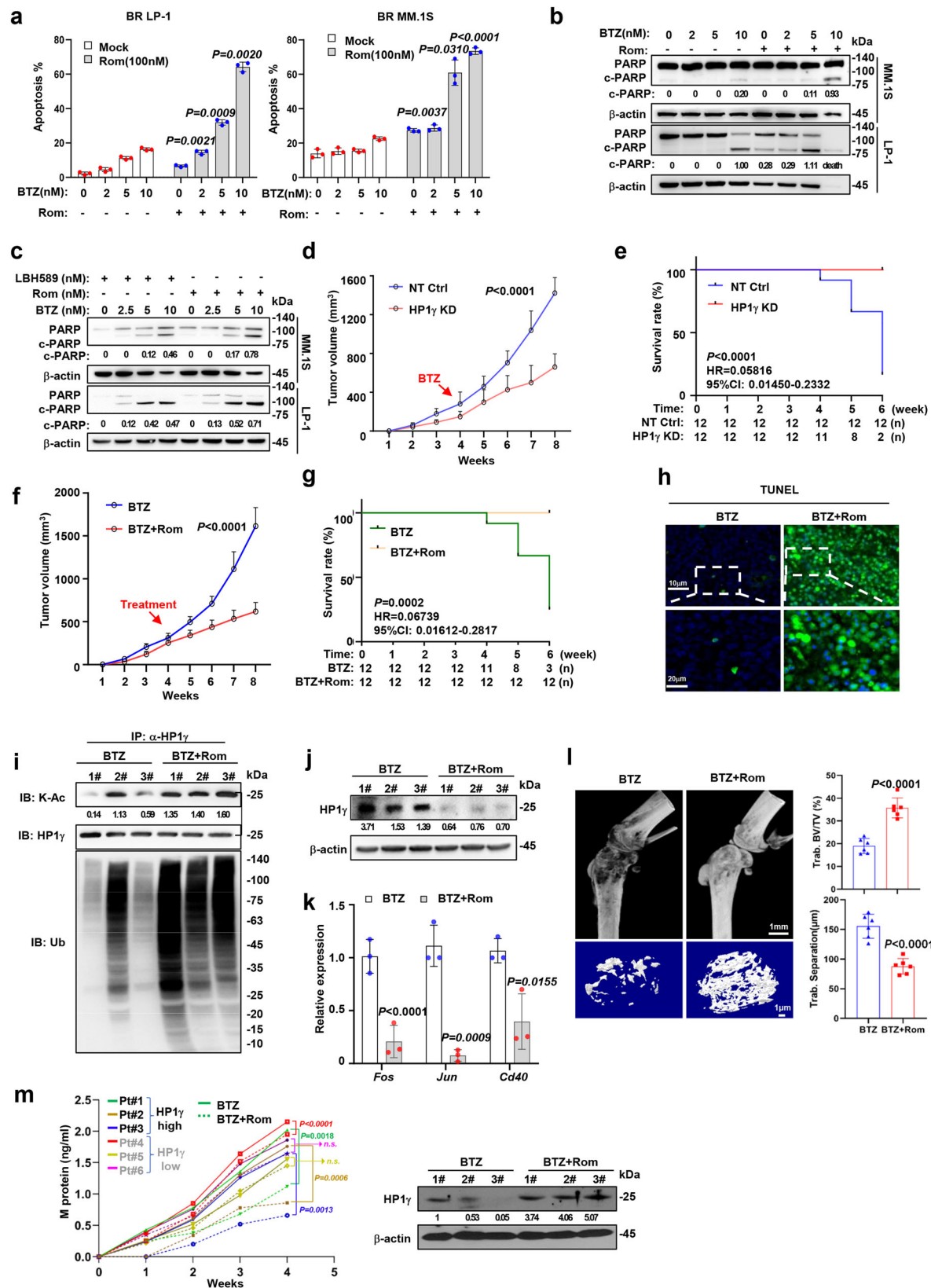

Guidelines of the World Medical Association Declaration of Helsinki. Signed informed consent was obtained from all participating individuals prior to participation in the study. Animal studies were approved by the Committee on Animal Research and Ethics of Tianjin Medical University. All protocols conformed to the Guidelines for Ethical Conduct in the Care and Use of Nonhuman Animals in Research.

## MM cell lines and primary CD138+ plasma cells

Myeloma cell lines LP-1 was purchased from ATCC (American Type Culture Collection, Manassas, VA, USA), MM.1S from the National Infrastructure of Cell Line Resource (Beijing, China), CAG from Chinese Academy of Sciences Cell Bank (Shanghai, China), respectively. MM cells were cultured in RPMI-1640 media supplemented with 15% of fetal

**Fig. 8 | HDAC1 inhibitor abolishes BTZ drug resistance and rehabilitates bone disruption in vitro and in vivo. a** Quantification of apoptotic BR MM cells treated with BTZ (0–10 nM) and 100 nM Rom for 24 h (mean ± s.d.; $n$ = 3 independent experiments). **b** Cleavage of PARP in BR MM cells treated with or without Rom, combined with increasing dosage of BTZ for 24 h. **c** Western blotting shows cleaved PARP in MM cell treated with increasing dosage of BTZ combined with LBH589 and Rom for 24 h. **d** Tumor growth of HP1γ KD or NT Ctrl BR MM.1 S cells (3×10⁶ cells/mouse, $n$ = 12/group) in NSG mice receiving BTZ (0.5 mg/kg) (mean ± s.d.). **e** Survival rate of mice at the time points of tumor diameter over 15 mm³. **f** Tumor growth of BR MM.1 S cells (3 × 10⁶ cells/mouse, $n$ = 12/group) in NSG mice receiving BTZ (0.5 mg/kg) or BTZ+Rom (1 mg/kg) (mean ± s.d.). **g** Survival rate of mice at the time points of tumor diameter over 15 mm³. **h** Immunofluorescence staining in tissues from xenograft of different mice groups. **i** Co-IP assay shows the ubiquitination and acetylation levels, and (**j**) protein levels of HP1γ in xenograft tissue of different treatment groups. #1–3: the mouse number selected randomly in BTZ and BTZ+Rom groups. **k** Relative expression of *Fos*, *Jun* and *Cd40* in xenograft tissue of different treatment groups (mean ± s.d.; $n$ = 3 independent experiments). **l** Left: representative microCT images of mouse femurs bearing BR MM.1 S cells and treated with BTZ or BTZ + Rom. Right: measurement of the percentage of bone volume to total volume (BV/TV) and trabecular number in the metaphyseal regions of the mice femurs (mean ± s.d.; $n$ = 12 mice/group). **m** Levels of M protein secreted in PDX mouse tail vein blood after treatment with BTZ or BTZ + Rom ($n$ = 3). Western blotting shows HP1γ-high and HP1γ-low level in the sorted CD138⁺ cell. $P$ values were determined by Pearson's coefficient and log-rank tests (**m**), one-side ANOVA (**a**, **d**–**g**), and Student's $t$ test (**k, l**). Source data are provided as a Source data file.

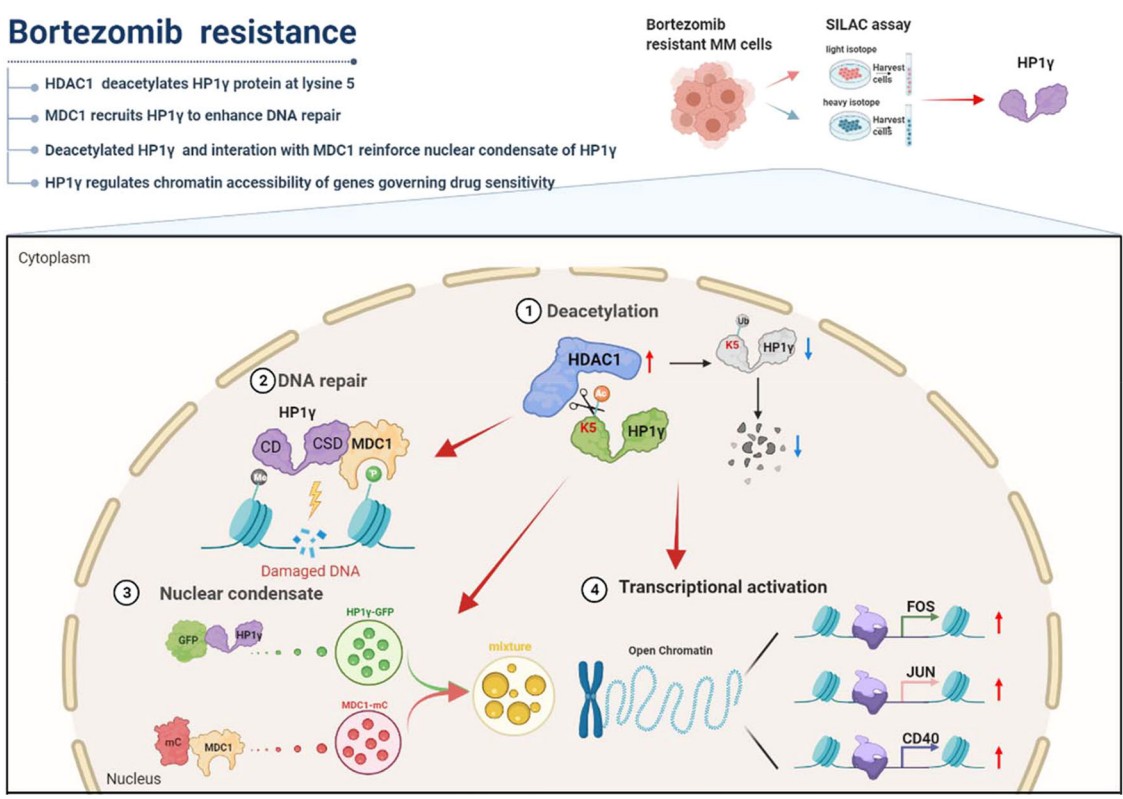

**Fig. 9 | Proposed schematic diagram for mechanisms of HP1γ-MDC1-HDAC1 regulating axis in MM drug resistance.** HP1γ-MDC1 complex recognizes the DSB sites and exerts the function of DNA repair, as well as how the nuclear condensation of HP1γ is regulated by deacetylation and facilitated by MDC1; HP1γ remodels histone methylation to facilitate accessibility of genes governing sensitivity to PIs; therefore, targeting HP1γ may be efficacious for overcoming drug resistance in MM patients.

bovine serum, 100 U/ml of penicillin, 100 mg/ml of streptomycin, and 2 mM L-glutamine (Gibco, Life Technologies, Carlsbad, CA, USA). The HEK 293 T and HEK 293 A cell was cultured in DMEM-high glucose media with 10% fetal bovine serum, 100 U/ml of penicillin, 100 mg/ml of streptomycin, and 2 mM L-glutamine. These cells were all cultured at 37°C in a humidified incubator with 5% CO₂ (Gibco, Life Technologies, Carlsbad, CA, USA).

CD138⁺ plasma cells were isolated using CD138 MicroBeads (Miltenyi Biotec, Bergisch Gladbach, Germany) from newly diagnosed or relapsed patients with multiple myeloma according to the manufacturer's instructions. Briefly, 3–5 ml of bone marrow biopsies were diluted up to 10 ml with RPMI-1640 media and gently loaded onto the top of 10 ml Ficoll Paque Plus (Sigma-Aldrich, St. Louis, MS, USA) (United States) and centrifuged at 800 × $g$ for 25 min at room temperature with the acceleration at 0. After centrifuge, the PBMCs were carefully aspirated from the Ficoll-plasma interface and transferred into a new 15 ml tube, and washed with PBS at 300 × $g$ for 10 min at room

temperature twice. For 2 × 10⁷ total cells, the pallet was resuspended in 80 μl buffer and labeled with 20 μl of CD138 MicroBeads at 4 °C for 15 min. After washed, the cells were resuspended in 500 μl buffer and proceeded for the positive selection of plasma cells from PBMC.

### Establishment of Bortezomib (BTZ)-resistant myeloma cells
To develop bortezomib-resistant (BR) myeloma cells, parental drug-naive cells were imitated by 0.5 nM of BTZ and enhanced by doubled dosage very one month up to six months totally. Acquire of BTZ-resistant phenotype were monitored and confirmed by calculating the IC₅₀ of BTZ using MTS assay. Cells with IC₅₀ over 10 times were kept for further experiments. For further details on the process of inducing drug resistance please refer to the literature[32].

### SILAC assay
The cells were labeled with either "heavy isotopic amino acids" (L-13C6-Lysine/L-13C615N4-Arginine) or "light isotopic amino acids" (L-

Lysine/L-Arginine) using a SILAC Protein Quantitation Kit (Pierce, Thermo) according to manufacturer's instructions. The cell line was grown for more than six generations before being harvested, to achieve more than 97% labeling efficiency. After that, the cells were further expanded in SILAC media to desired cell number. Finally, the cells were washed with cold PBS two times and store at −80 °C for further LC-MS/MS and bioinformatics analysis (PTM biolabs, Hangzhou, China).

### Transfection, virus package and infection
Transient transfections to HEK 293 T cells were performed using polyethyleneimine (PEI) (Polysciences, Warrington, PA, USA) in the OPTI-MEM medium (Life Technologies, Carlsbad, CA, USA) with a ratio of 1:4 to 1:6 of DNA: PEI.

Viral particles were produced by HEK 293 T cells in a 10 cm dish transfected with 4 μg pMD2.G and 6 μg psPAX2 packaging plasmids (Addgene, Watertown, MA, USA), together with 8 μg lentiviral expressing vectors encoding target genes, including pITA -CBX3-N/C-flag, pLKO.1 or hU6-MSC-Ubiquitin-eGFP vector encoding shRNAs targeting interested genes as listed in the Supplementary Information. Supernatant carrying the viral particles was harvested 35 h and 60 h after transfection and concentrated to 100× volume by Poly (ethylene glycol) 8000 (Sigma-Aldrich, St. Louis, MS, USA).

For viral infection, $1 \times 10^6$ myeloma cells were seeded in 1 ml new complete media for 6 h and then added 50 μl viral concentration and 8 μg/mL polybrene, and cells were spin at $800 \times g$ for 45 min at 20 °C. In total, 12 h after spinfection, the medium was changed and cells were cultured for another 48 h until further management.

### Immunoassays
Antibody-based immunoblotting has been described in our previous study[16,37]. All antibodies, venders, dilutions were provided in the Supplementary Information. The representative Western blot images for at least three independent experiments shown in the figures have been cropped and auto contrasted.

### Cell Proliferation assay
Myeloma cells ($5 \times 10^3$) infected with lentivirus carrying the CBX3-flag (CBX3 OE) or vector control (Vector) were treated as designed in 96-well plates for appropriate time and then added 20 μl of CellTiter 96 AQueous One Solution Reagent (Promega, Fitchburg, WN, USA) into 100 μl of culture media, after incubation for 1–4 h at 37 °C the plates were read at 490 nm with a Microplate Reader 550 (Bio-Rad Laboratories, Richmond, CA, USA). The following formula was used to calculate cell viability (%) = OD value of treatment group/OD value of control group ×100.

### RNA extraction and real-time PCR
Total RNA was isolated using Trizol (Life Technologies, South San Francisco, CA USA) according to the manufacturer's instructions. Total RNA (2 μg) was reverse transcribed using the 5× All-In-One reverse transcription MasterMix (abm, Vancouver, Canada). Quantitative real-time PCR was performed by mixing cDNA, gene-specific primers and EvaGreen 2× qPCR MasterMix (abm, Vancouver, Canada) in the QuantStudio 3 Real-Time PCR System (Applied Biosystems). Primer sequences are provided in the Supplementary Information.

### RNA-seq experiment
Myeloma cells were infected with lentivirus carrying CBX3 OE or Vector in complete medium treated and subsequently were verified its overexpression effect by Western blot. RNA was isolated using Trizol (Life Technologies, South San Francisco, CA USA) according to the manufacturer's instructions. Quality of the purified RNA was tested on Agilent 2100 Bio analyzer (Agilent RNA 6000 Nano Kit) (BGI, Shenzhen, China). Libraries for cluster generation and DNA sequencing were prepared following an adapted method from BGISEQ-500 platform. The low-quality reads (more than 20% of the bases qualities are lower than 10) were filtered to get the clean reads. Then those clean reads were assembled into Unigenes, followed with Unigene functional annotation, SSR detection and calculate the Unigene expression levels and SNPs of each sample. Finally, DEGs (differential expressed genes) were identified between samples and do clustering analysis and functional annotations.

### ATAC-seq experiment
Cells ($5 \times 10^4$) were spun at $500 \times g$ for 5 min, followed by a wash using 50 μL of cold 1× PBS and centrifugation at $500 \times g$ for 5 min. Cells were lysed using cold lysis buffer (10 mM Tris-Cl, pH 7.4, 10 mM NaCl, 3 mM MgCl2 and 0.1% IGEPAL CA-630). Immediately after lysis, nuclei were spun at $500 \times g$ for 10 min using a refrigerated centrifuge. To avoid losing cells during the nuclei preparation, we used a fixed angle centrifuge and carefully pipetted away from the pellet after centrifugations. Immediately following the centrifugation, the pellet was resuspended in the transposase reaction mix (10 μL 5× TTBL buffer, 3 μL TTE Mix V50 (Illumina) and 37 μL of nuclease free water). The transposition reaction was carried out for 30 min at 37 °C. Directly following transposition the sample was purified using a DNA Clean&Concentrator-5 kit (ZYMO RESEARCH, D4014, California, USA). Following purification, we amplified library fragments using Q5® Hot Start High-Fidelity DNA Polymerase and PCR primers N5 and N7 (TruePrep® Index Kit V2 for Illumina, Vazyme, TD202), using the following PCR conditions: 72 °C for 5 min, 98 °C for 30 s, followed by thermocycling at 98 °C for 10 s, 63 °C for 30 s and 72 °C for 1 min. We amplified the full libraries for 13 cycles, after 13 cycles we took purification to the PCR reaction by VAHTS DNA Clean Beads (Vazyme, N411-01) according to the manufacturer's instructions. Purified DNA was analyzed by high-throughput sequencing (Novogene, Beijing, China).

### NHEJ assay and HR assay
A total of $2 \times 10^6$ HEK 293 T cells transduced with vector, HP1γ-WT or K5R were transfected with 1 μg of eGFP-NHEJ/HR reporter with RFP vector as control. Cells were harvested 48 h later and assayed with flow cytometry system (BD Bioscience) for GFP expression. Please refer to the literature for specific experimental details[33].

### Apoptosis assay
After processing accordingly of MM cells for the indicated time, apoptosis assay was then carried out using the Annexin V-FITC Apoptosis Detection Kit (Sigma-Aldrich, St. Louis, MO, USA) according to the manufacturer's instructions. A total of $1 \times 10^5$ cells were stained with 5 μl Annexin V-FITC and 1 μl of PI in dark. The cells were analyzed by FACS Calibur instrument. The data of flow cytometry was analyzed with CellQuest 3.0 software (BD Biosciences, New Jersey, USA). Cells were stained negatively with Annexin V and PI were considered viable cells, and early apoptotic cells were positive for Annexin V and negative for PI, and late apoptotic cells were positive for Annexin V and PI.

### Cell cycle analysis
After the corresponding cells were treated, they were harvested and resuspended at $0.5–1 \times 10^5$ cells/ml in fixing solution. Before being analyzed by flow cytometry, cells were resuspended in staining buffer, treated with 25 μL RNase A and stained with 10 μL propidium iodide (PI) (Beyotime, Shanghai, China). Then, the cells were analyzed by FACS Calibur instrument and the data of flow cytometry was analyzed with CellQuest 3.0 software (BD Biosciences, New Jersey, USA).

### Laser micro-irradiation
The CAG and HEK 293 A cells were infected with MDC1 KD or NT Ctrl. Laser micro-irradiation was performed using the PLAM Micro Beam

(ZEISS, oberkochen, German). The laser output was set to 45%, which can reproducibly give a focused γH2AX stripe. Then, the samples were subjected to immunofluorescence staining with indicated antibodies.

## Immunofluorescence staining

The corresponding cells were fixed in 4% formaldehyde for 10 min, then samples were treated in 0.5% (V/V) Triton X-100 for 15 min and blocked with 5% BSA for 30 min at 37 °C. After incubated with anti-HP1γ antibody (1:100) (Cell Signaling Technology, #2619, Danvers, MA, USA) and anti-γH2AX antibody (1:100) (millipore, 05-636, Darmstadt, Germany) overnight at 4 °C, samples were incubated with Alexa Fluor® 555 donkey anti-rabbit IgG (H + L) or Alexa Fluor® 555 donkey anti-rabbit IgG (H + L) (1:2000) for 60 min at room temperature and nucleus counterstaining with DAPI. Imaging was obtained by the Olympus FV1000 IX81-SIM Confocal Microscope (Olympus, Tokyo, Japan).

## Tunnel assay

Tunnel assay was performed according to DeadEnd™ Fluorometric TUNEL System (Promaga, Tokyo, Japan) manufacturer's instructions. Briefly, tumor sections were treated with fresh xylene in a Coplin jar for 5 min twice at room temperature. Rehydrate the samples by sequentially immersing the slides through graded washes (100%, 95%, 85%, 70%, 50%, 0.85% NaCl, PBS) for 5 min. The tissue sections were fixed by immersing the slides in 4% methanol-free formaldehyde solution in PBS for 15 min. After washed in PBS for 5 min three times, the samples were incubated with 100 μl proteinase K (20 μg/ml) for 8–10 min. Then were washed and incubated with 100 μl Equilibration Buffer for 5-10 min. The tissue sections were fixed with 4% methanol-free formaldehyde solution in PBS for 5 min. Then washed and incubated with 50 μl of rTdT incubation buffer at 37 °C for 60 min in the dark. The reactions were terminated by incubated with 2X SSC for 15 min at room temperature. The samples were washed three times and were stained by propidium (1 μg/ml) in PBS for 15 min at room temperature in the dark. Then the samples were washed three times and analyzed by the Olympus FV1000 IX81-SIM Confocal Microscope (Olympus, Tokyo, Japan).

## Fluorescence recovery after photobleaching (FRAP)

The HEK 293 A transfected with correlated plasmids and the relevant purified protein were used for FRAP experiments with an Olympus FV1000 IX81-SIM Confocal Microscope (Olympus, Tokyo, Japan). Photobleaching was performed using tornado mode with the 488 nm laser at 45% laser power for GFP. Fluorescence recovery was monitored with 488 nm laser using the free-run mode at 1 s-4.2 s interval. Fluorescence of unbleached site in the same view was also monitored as the control. Signal was presented as the ratio relative to the fluorescence signal before photobleaching.

## In vitro droplet assay

For protein purification, Plasmids containing the protein of interest fused to GFP were transformed into BL21 cell (NEB, C2527I). A fresh bacterial colony was inoculated into 200 ml LB media containing kanamycin and grown at 37 °C until an OD$_{600}$ of 0.8–0.9 has been reached. IPTG was added to 1 mM and growth continued overnight at 16 °C. Pallets from 200 ml cells were resuspended in 10 ml of GST lysis buffer (50 mM Tris-HCl pH7.5, 100 mM NaCl) containing 1 mM dithiothreitol, 0.2 mM Phenylmethylsulfonyl fluoride, 1% Triton-X100, complete protease inhibitor and sonicated (4 cycle of 30 sec on, 30 sec off). The lysate was cleared by centrifugation at 12,000 × g for 20 min at 4 °C and added NaCl to 500 mM. Then the supernatant was added 500 μl glutathione agarose (Thermo Fisher, 16100) (prewashed in lysis buffer). Tubes containing this agarose lysate slurry were rotated at 4 °C overnight. Then the packed agarose washed twice with GST lysis buffer containing 500 mM NaCl and twice with GST lysis buffer. Protein was

obtained by cleavage of Pierce™ HRV 3 C protease and judged by coomassie stained gel.

For droplet assay, protein was added to solutions at varying concentrations with indicated final salt and molecular crowder concentrations in Buffer A (50 mM Tris-HCl pH 7.5, 10% glycerol, 1 mM DTT). The protein solution was immediately loaded onto a homemade chamber comprising a glass slide with a coverslip attached by two parallel strips of double-sided tape. Slides were then imaged with an Andor confocal microscope with a ×100 objective. Imaging was obtained by the Olympus FV1000 IX81-SIM Confocal Microscope (Olympus, Tokyo, Japan).

## Turbidity assay

This assay measures the turbidity of the phase-separated solution by absorbance at 600 nm with NanoDrop (Thermo scientific). Protein samples were prepared by mixing determined amounts of targeted protein, NaCl, and buffer to achieve desired concentrations of each component. Absorbance at 600 nm was monitored and recorded at room temperature by NanoDrop. Absorbance values were reported after subtracting the optical density of buffer. Data were collected in duplicate (some are $n = 10$) and representative traces are presented.

## Western blotting

Protein lysates were prepared in RIPA-buffer (50 mM Tris-HCl pH 7.5, 150 mM NaCl, 10 mM EDTA, 0.5% sodium deoxycholate, 1% NP-40, 1 mM sodium dodecyl sulfate, 10 μg/mL aprotinin, 1 mM phenylmethanesulfonyl fluoride, and 10 μg/mL leupeptin) supplemented with complete protease inhibitors (Roche, Indianapolis, IN, USA). The protein concentration was determined using the BCA protein assay kit (ThermoFisher Scientific, Carlsbad, CA, USA). Cell lysate (50 μg) was separated by electrophoresis on SDS-PAGE gel and transferred to nitrocellulose membranes (Pall Corporation, Washington, NY, USA). Membranes were blocked with 5% non-fat milk for 1 h at room temperature and infiltrated overnight at 4 °C with specific antibodies. Antibodies used in this study were listed in the Supplementary Information. Membranes were washed three times in PBST the next day, then incubated with horseradish peroxidase-conjugated secondary antibodies for 1 h at room temperature, washed three times with PBST and finally bands were visualized using an enhanced chemiluminescence system (Millipore, Los Angeles, CA USA). The representative western blot images for at least three independent experiments shown in the figures have been cropped and auto contrasted.

## Co-immunoprecipitation (Co-IP)

Cells were harvested and lysed by NP-40 lysis buffer (50 mM Tris-Hcl pH 7.4, 150 mM NaCl) supplemented with complete protease inhibitors (Roche, Indianapolis, IN, USA) on ice for 30 min. Cell lysate was centrifuged for 20 min at 12,000 × g at 4 °C. Co-IP for exogenous expressed proteins, supernatant was incubated with anti-FLAG M2 Affinity Gel (Sigma-Aldrich, St. Louis, MO, USA) at 4 °C overnight. Other antibodies used in this study were listed in the Supplementary Information. The next day, the pellet was washed four times with NP-40 lysis buffer, and then subjected to western blotting analysis using the anti-acetylated lysine or anti-GFP antibodies, respectively. Mass spectrometry data is provided in Supplementary Data 3.

## Chromatin-immunoprecipitation (ChIP) and ChIP-sequencing (ChIP-seq)

$4 \times 10^7$ cells were washed in PBS and cross-linked with 1% formaldehyde for 10 min at room temperature and then quenched by addition of glycine (125 mM final concentration) for 5 min. For Nuclei isolation, cells were resuspended in cell lysis buffer (50 mM Tris pH8.0, 140 mM NaCl, 1 mM EDTA, 10% glycerol, 0.5% NP-40, 0.25% Triton X-100), incubated the tube on ice for 20 min to swell. Harvested the nuclei by centrifugation at 2000g for 5 min at 4 °C resuspended in 1 ml ChIP lysis

buffer (1% SDS, 10 mM EDTA, 50 mM Tris-HCl, pH8.0) and incubated on ice for 10 min. Chromatin was fragmented to 200–500 bp using 12 cycles using the Vibra-Cell Ultrasonic Liquid Processors (SONICS, Newtown, CT, USA). For each IP, chromatin was immunoprecipitated with 2 mg of antibody in IP dilution buffer (1% Triton X-100, 2 mM EDTA, 150 mM NaCl, 20 mM Tris-HCl, pH 8.0) at 4 °C overnight. Chromatin was precleared for 2 h each with protein G agarose beads (Cell Signaling Technology, Danvers, MA, USA) before immunoprecipitation. The immunoprecipitated material was washed, once in TSE I buffer (20 mM TrisHCl pH 8.0, 2 mM EDTA pH8.0, 150 mM NaCl, 1% Triton X-100, 0.1% SDS), once in TSE II buffer (20 mM TrisHCl pH 8.0, 2 mM EDTA pH8.0, 500 mM NaCl, 1% Triton X-100, 0.1% SDS), once in LiCl buffer (10 mM TrisHCl pH 8.0, 250 mM LiCl, 1% deoxycholic acid, 1% NP40) and once in TE buffer (10 mM Tris pH 8.0, 1 mM EDTA pH8.0) before elution in elution buffer (100 mM NaHCO3, 1% SDS). Antibodies used in this study were listed in the Supplementary Information. The samples were removed from beads, reversed cross-linked overnight at 65 °C and DNA was isolated using QIAquick PCR Purification Kit (Germantown, MD, USA). Precipitated DNA was analyzed by high-throughput sequencing (Beijing Genomics Institute, Beijing, China).

## Immunohistochemistry
Deparaffinize myeloma tissue array with normal bone marrow tissue slides in xylene for 2 times, 15 min each. Transfer slides to 100% alcohol, for 2 times, 5 min each, and then transfer once through 95%, 70 and 50% alcohols sequentially for 5 min each. Block endogenous peroxidase activity by incubating sections in 3% $H_2O_2$ solution at room temperature for 10 min to block endogenous peroxidase activity. Rinse with PBS twice, 5 min each. Pour 10 mM citrate buffer pH 6.0 into the staining container and incubate it at 98 °C for 20 min. Remove the staining container to room temperature and allow the slides to cool for 40 min. Rinse slides with PBS for 2 times, 5 min each. Add blocking buffer onto the sections of the slides and incubate in a humidified chamber at room temperature for 1 h. Drain off the blocking buffer from the slides. Apply appropriately diluted primary antibody to the sections on the slides and incubate in a humidified chamber at 4 °C overnight. Wash the slides with PBS for 3 times, 5 min each. Apply appropriately diluted biotinylated secondary antibody to the sections on the slides and incubate in a humidified chamber at room temperature for 1 h. Wash slides with PBS for 3 times, 5 min each. Apply DAB substrate solution (Dako, K5361) (freshly made just before use) to the sections on the slides to reveal the color of antibody staining. Allow the color development for less than 10 min until the desired color intensity is reached. Wash slides with PBS. Counterstain slides by immersing sides in Hematoxylin for 1–2 min. Rinse the slides in running tap water for 10 min. Dehydrate the tissue slides through 5 times of alcohol (50%, 70%, 95%, 100 and 100%), 1 min each. Clear the tissue slides in 2 times of xylene and coverslip using mounting solution. The mounted slides can be ready for photo capture or stored at room temperature permanently.

## NOD/SCID xenograft and bone lesion mice models
Animal studies were approved by the Committee on Animal Research and Ethics of Tianjin Medical University, and all protocols conformed to the Guidelines for Ethical Conduct in the Care and Use of Nonhuman Animals in Research. 4–6 weeks old female NOD.$Cg$-$Prkdc^{scid}Il2rg^{tm1Wjl}$/ SzJ (NSG) mice were used to establish the xenograft and intra-bone injection models as previously reported[38,39]. The mice were bred and maintained under specific pathogen-free conditions at the animal facilities of Tianjin Medical University. These mice were maintained in 12 h light/dark cycle, and the housing temperature and humidity were 21.5–24.5 °C and 45–65%, respectively. Mice were given standard laboratory chow diet and water ad libitum. For Xenograft model, MM cells ($3 \times 10^6$ cells/mouse) were injected subcutaneously into NSG mice. After 3 weeks, mice were treated with BTZ (0.5 mg/kg) ($n = 12$)

every three days or BTZ + Romidepsin (1 mg/kg) ($n = 12$) every two days. Mice were weighted and tumors were measured every 3 days. After treatment 24 days, for xenografts experiments, mice were sacrificed and the tumor xenografts were collected for immunohistochemistry (IHC) and apoptosis analysis. The tumor volumes of all tumor-bearing mice involved in this study were controlled within 2500 mm³, which is in accordance with the permission from the ethics committee of Tianjin Medical University. The maximal tumor size/ burden was not exceeded. For intra-bone injection experiments, BR MM.1 S ($5 \times 10^5$/mouse) were injected into the femurs of NSD mice. Mice without myeloma cells served as controls (No MM). The frequency and dose of drug injection in mice are consistent with the subcutaneous experiment. Mice femur were subjected to microCT scan with a Skyscan 1172 microtomograph, and mouse femurs were subjected to histological evaluations, where shows representative microCT reconstructions of mouse femurs to show osteolytic lesion area and less cortical perforations.

## X-ray microtomography
MicroCT was performed on mice femur with a Skyscan 1172 microtomograph (BrukermicroCT, Kontich, Belgium). After segmentation, the 3D models were constructed from the stack of 2D images with a surface-rendering program (Ant, release 2.0.5, Skyscan). 3D measurements were obtained with the CtAn software (release 2.5, Skyscan). Trabecular bone analysis was performed on the femur body. The following 3D parameters were calculated: trabecular volume (BV/TV, in%) and trabecular number (Tb.N, in mm⁻¹).

## Droplet detection and data analysis
In vivo puncta of phase separation were identified according to previous reports[13,40–42]. Images for each channel (blue, green, and red) were exported as individual non-compressed TIFF files by the built-in FV10-ASW viewer software and individually analyzed by Photoshop. For image analysis, only the Brightness/Contrast of the entire image was adjusted manually, and the parameters were kept the same for the same batch of images under the same channel (different channels may be adjusted differently). To preclude experimental variations, only puncta for HP1γ or MDC1 in images obtained from the same experiments that have been applied with the same brightness/contrast adjustment were analyzed. For presentation, representative raw images were adjusted in brightness and contrast linearly and equally within the samples using ImageJ software. Image merging and cropping were done in Photoshop. Scale bar was applied to individual images before cropping. The puncta size for HP1γ (wild type or over-expressed mutants) and MDC1 were analyzed by ImageJ software based on the standard procedures. Particle analysis was used in ImageJ to calculate the fluorescent intensity and area of puncta condensates in each sample, and we applied the script to all raw image sets in batch mode. In the script, a Watershed Separation and background correction was applied to each image, followed by particle analysis. We measured all images after converting the image type, correcting the optical density, adjusting and fixing the image threshold, separating particles using Watershed (Process-Binary-Watershed), and analyzing particles in ImageJ software (Please refer to more detailed procedure from Particle Analysis (imagej.net) and Particle Analysis - ImageJ). Each parameter was determined from particle analysis for endogenous HP1γ in WT MM.1 S cells. The exported data table was further summarized in Excel and GraphPad software. According to the sizes and pixels of the original figure, we calculated the sizes of particles. For HP1γ or MDC1 puncta formed, we determined their size to be 1.057 μm² (median in the area) with 20 and 95% percentiles to be 0.523μm² and 5.178μm², respectively. Hence, we counted those dots with area ranging 0.523 μm² to 5.178 μm². Similarly, we measured the co-location size for the yellow puncta formed by HP1γ and MDC1, and defined the puncta with the same standard of 20 and 95% percentiles.

## Statistics and reproducibility

Statistical details of experiments can be found in the figure legends. All data were shown as mean ± SD for at least three independent experiments. In vitro experiments were determined using paired two-tailed Student's $t$ test or Mann–Whitney nonparametric test, and two-way ANOVA plus Bonferroni post hoc test were used for in vivo experiments. Pearson correlation test was used to determine the correlations between gene expressions. A $p$ value less than 0.05 was considered statistically significant.

## Reporting summary

Further information on research design is available in the Nature Portfolio Reporting Summary linked to this article.

## Data availability

The RNA-seq and ATAC-seq data can be publicly found at the Gene Expression Omnibus database under accession number GSE176547. The Multiple Myeloma Research Foundation publicly available data used in this study are available in the Gene Expression Omnibus database under accession code GSE2658 and GSE9782. The source data of proteomics can be found in the database iProX for project IPX0005951000 with ID number PXD040298 [https://www.iprox.cn//page/project.html?id=IPX0005951000]. Source data are provided with this paper. Requests for any materials in this study should be directed to Zhiqiang Liu and obtained through an MTA. Source data are provided with this paper.

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

## Acknowledgements

This work was supported by the Beijing Natural Science Foundation of China (Z200020), the National Natural Science Foundation of China (81870161, 82070221, Z.L.; 81900215, J.Y. Wang; 82000216, Q.L.; 82270208, YF.W.), and the Tianjin Research Innovation Project for Postgraduate Students (2020YJSB162, Y.X.; 2021YJSB277, HM.J.).

## Author contributions

X.L. and ZQ.L. contributed to writing the manuscript; HM.J., Q.L., YF.W., J.G., YX.W., ZY. P., JY.W., and MQ.W. contributed to performing the experiments and statistical analyses; X.L., Y.X., and S.W. were in charge of the animal studies; ZQ.L. contributed to the design of the experiments.

## Competing interests

The authors declare no competing interests.
