## [Peer Review File · Nature Communications]

Deacetylation Induced Nuclear Condensation of HP1 γ Promotes Multiple Myeloma Drug ResistanceThis manuscript has been previously reviewed at another journal that is not operating a transparent peer review scheme. This document only contains reviewer comments and rebuttal letters for versions considered at *Nature Communications*.

REVIEWER COMMENTS

Reviewer #1 (Remarks to the Author):

In this article, Li et al, used in vitro proteomics approach to reveal the contribution of HP1 γ in resistance of multiple myeloma patients to bortezomib, a proteasome inhibitor. Authors have shown that HDAC1-mediated deacetylation of HP1 γ at lysine 5 residue, increases the HP1 γ condensation in the nucleus and leads to the recruitment of HP1 γ by MDC1 to induce DNA damage repair and enhances chromatin accessibility at the promoter of its target genes that regulate the response to bortezomib. Using in vitro and in vivo models, authors suggested that combination of HDAC inhibitor romidepsin and bortezomib could overcome the bortezomib resistance in MM.

We thank the authors for addressing our comments. We have a few more requests, most of which are related to data or answers presented following revisions. Please, see below:

Authors did intersect HP1 γ ChIP-seq and RNA-seq datasets to identify the molecular mechanism of HP1 γ -mediated bortezomib resistance and found 272 genes in common. However, authors did not dig into the ChIP-seq data that could be a very important source for defining the contribution of HP1 γ in DNA repair and bortezomib resistance in MM cells. How many genes or regions were differentially bound by HP1 γ in BR and control cells? What were the GO groups of HP1 γ target genes in BR and Ctrl cells?

Authors showed that majority of HP1 γ peaks were located within the non-coding regions (Fig. 6D) and they showed changes in the abundance of H3K36me3 and H3K9me3 marks at the promoter of those genes. Were the results confirmed in ATAC-seq dataset? How did genome-wide differential abundance of HP1 γ correlate with differential gene expression in RNAseq data and eventually in resistance to bortezomib?

Authors only selected three HP1 γ target genes due to their roles in DNA repair and drug resistance based on previous studies. It would have been nice if in a more unbiased approach, the critical genes would have been discovered and validated.

Contribution of HP1 γ expression and stabilization in response to bortezomib have been tested in vitro and in vivo. Authors have shown that depletion of HP1 γ or its destabilization using HDAC inhibitors affect viability of tumor cells. Given the genome-wide impact of HDAC inhibition, authors did not exclude the off-target effects of those drugs on treated cells.

Some of the labelling of the figures and figure legends throughout this manuscript do not correspond to what is written in the main text. It was very confusing to read and understand what data is being described in in each figure (no Supplementary figure 7, Fig. 4h, Fig. 3M and 3N).

Quality of Supplementary figures was very low, and it was very hard to read the figures.

Line 279: MSC1 instead of MDC1.

Reviewer #2 (Remarks to the Author):

Overall, I praise the effort by Li et al to build on the constructive feedback by all three reviewers. I believe the manuscript improved, particularly in addressing the functional link between HP1 γ -linked molecular actuators (JUN, FOS, CD40) and drug resistance (e.g., Fig. S8H-J). I also appreciate the effort to add quantifications to the IF and Western Blots. That said, however, I still find issues with the overall rigor and mechanistic interpretation of the HP1 γ condensation data. As detailed below, I urge the authors to address these remaining major concerns thoroughly before publication:

Major-1: The interpretation and quantitative analysis of intracellular HP1gamma (and MDC1) puncta (e.g., Fig. 4C, Fi. S8H-J) are largely obscured by the lack of a proper definition of what they consider intracellular puncta. This issue is complex, and should be addressed/acknowledged, given that the aggregation patterns are so dissimilar across the range of cells and treatments. The quality of Fig. S8H-I images is too low for future readers to judge puncta. For example, I can barely tell the difference between MM1.5 +/- Rom for the MDC1 staining; yet quantifications look striking. Crucially, the new provided methods misses the mark in helping understand how they segment and count puncta. Moreover, quantifications usually show 3 data points, but I would expect to see many data points (wherein each data point corresponds to number of puncta per cell) for each condition (i.e., for each bar; please add data for all quantified cells).

Major-2: Adding to point (1), the conclusions of the study largely rely on establishing a connection between HP1gamma puncta and MDC1 puncta in the relevant cell types. I previously requested quantification of MDC1 enrichment in HP1gamma puncta for the relevant drug resistant cell types. The authors didn't add these crucial data. The new Fig. S8H-J data fail to link MDC1 and HP1gamma at the puncta level. They should study/quantify colocalization of the MDC1 and Hp1 gamma signals.

Major-3: Totally unclear how data in Fig. S7I (related to statement in lines 277-278) show a reduction in puncta upon HDAC inhibition (+Rom). It isn't obvious from the IF images and the quantifications are not labeled --if read as other unlabeled quantifications where the trends are obvious, the data would suggest increase in puncta upon Rom treatment. Other examples of bar quantification without proper labels: Fig. 4C.

Major-4: I remain concerned by the insistence (e.g., lines 375-376) on the model that HP1gamma condensates recruit MDC1. The authors never quantify the colocalization of MDC1 and HP1gamma. They also did not study MDC1 gamma recruitment to sites of DNA damage under HP1gamma KD -- despite having the tools to do that. Curiously, the subsequent discussion in lines 381 to 388 describes a model that puts MDC1 upstream of HP1gamma, as their data do support. The underlying discrepancy needs to be corrected in favor of what their data show --unless they add new data to suggest that HP1gamma indeed recruits MDC1.

Major -5: regarding the contribution to clinical translation, the authors argue that their study helps "define [the] applicable population" for combination therapy. They should add clarity to this statement, namely, what is the applicable population based on the new findings?

Minor:

- Fig S8 and S9 are mislabeled (should be s7 and s8)
- Fig. S8J, what is DMS? DMSO?
- Fig. 4I, note that the X axis label for "40" is misplaced
- Fig. 4H-I is presented as evidence that HDAC1-mediated deacetylation enhances LLPS, but the figure is mislabeled as "acetylation" for the treated group.
- What is the control in 4J? (vector?). The use of "vector" throughout the manuscript is often imprecise. If they mean "empty vector", then say so.
- Lines 206-208. "Moreover, 206 in MM cells, when the HDAC1 was suppressed by its inhibitor Rom, or forcedly expressed, the nuclear condensation of HP1 γ was remarkably inhibited or promoted, respectively (Fig. 4C)." Fig. 4C shows OE, so missing SI citation.
- Line 210 "above proved that K5" should be "probed K5"
- Line 220-221 incorrectly states that the data for FL corresponds to HEK293 in vivo; when they seem to correspond to an in vitro assay --and the HEK data are for the IDR1 variants
- Line 279. Note "MSC1" puncta instead of "MDC1"
- Line 368: "new machinery" is not the proper term. New mechanism?
- Line 370: should be "which may reinforce". The authors do not provide convincing data to show that HP1gamma condensation recruits MDC1 to sites of DNA damage --they do provide data to show that MDC1 at sites of DNA damage is required for HP1gamma recruitment. I pointed this out before, and the authors largely corrected related statements, except here.
- Lines 421: "trails" should be "trials". The authors should also cite the corresponding literature.

Reviewer #3 (Remarks to the Author):

the authors have sufficiently addressed my comments to present an improved manuscript. I think this new version of the manuscript is a valuable contribution to the literature.

RESPONSE TO REVIEWERS' COMMENTS

Reviewer #1 (Remarks to the Author):

In this article, Li et al, used in vitro proteomics approach to reveal the contribution of HP1 γ in resistance of multiple myeloma patients to bortezomib, a proteasome inhibitor. Authors have shown that HDAC1-mediated deacetylation of HP1 γ at lysine 5 residue, increases the HP1 γ condensation in the nucleus and leads to the recruitment of HP1 γ by MDC1 to induce DNA damage repair and enhances chromatin accessibility at the promoter of its target genes that regulate the response to bortezomib. Using in vitro and in vivo models, authors suggested that combination of HDAC inhibitor romidepsin and bortezomib could overcome the bortezomib resistance in MM.

We thank the authors for addressing our comments. We have a few more requests, most of which are related to data or answers presented following revisions. Please, see below:

Authors did intersect HP1 γ ChIP-seq and RNA-seq datasets to identify the molecular mechanism of HP1 γ -mediated bortezomib resistance and found 272 genes in common. However, authors did not dig into the ChIP-seq data that could be a very important source for defining the contribution of HP1 γ in DNA repair and bortezomib resistance in MM cells. How many genes or regions were differentially bound by HP1 γ in BR and control cells? What were the GO groups of HP1 γ target genes in BR and Ctrl cells?

Response: Firstly, we greatly appreciate your evaluation of our revision work, and also thank you for your insightful comments. We agree with the reviewer that in-deep analysis of ChIP-seq of HP1 γ is a very important source for defining key genes governing DNA repair and chemoresistance in MM cells, and we actually did use this strategy in this study. Since HP1 γ is a reader protein binding genome-wide chromatin methylated at H3K9 (Ragnhild Eskeland, doi: [10.1128/MCB.01576-06](https://doi.org/10.1128/MCB.01576-06)), not like transcriptional factors binding some certain specific genes, thus the immunoprecipitated genes either in WT or in BR MM cells were tremendous. We analyzed the ChIP-seq data in the WT and BR MM cells, and identified 17,246 bound genes in the BR cells and 10,023 bound genes in the WT cells. Using the MAnorm tool (Genome Biol. 2012 Mar 16;13(3): R16. doi: [10.1186/gb-2012-13-3-r16](https://doi.org/10.1186/gb-2012-13-3-r16)), we identified 5,064 overlapped genes,

7,925 differentially bound genes (DBG) in the BR groups, and 1,9793 DBG genes in the WT cells. Importantly, when these genes were intersected with upregulated genes in the RNA-seq data of BR cells, we identified 188 genes that were upregulated in the BR cells. Then, we validated expressions of 25 genes that have been reported to act in chemoresistance or DNA repair, and found that expressions of *FOS*, *JUN* and *CD40* were among the top ones.

Per your questions, the numbers of genes in the WT and BR MM cells immunoprecipitated by HP1 γ , and the KEGG analysis of the intersected genes were summarized in the new S Figure 8:

The GO enrichment of the DBGs of BR cells showed very general biological processes, cellular components, and molecular functions were enriched, because too many genes were gathered. Since this analysis is not as specific as the KEGG result, we did not show it in the main text and only showed it to answer this question. Actually, we have upload all sequencing data onto the Gene Expression Omnibus database under accession number [GSE176547](https://www.ncbi.nlm.nih.gov/geo/query/acc.cgi?acc=GSE176547), anyone who are interested is able to analyze according to their own strategy.

The related figures are revised in Figure 6:

And the related descriptions were also revised in the main text:

Our ChIP-seq of HP1 γ showed 17,246 and 10,023 genes in the BR and WT MM cells, respectively (Fig. S8A), and analysis of the differently bound genes (DBG) in the BR cells showed replication and repair, transcription, signaling molecules and transduction, cancer drug resistance processes were enriched by KEGG analysis (Fig. S8B). Integrated with RNA-seq data, 188 genes were intersected in the DBGs and HP1 γ upregulated genes (Fig. 6E, Table S1). Of these intersected genes, we evaluated 25 of them in the BR MM cells compared with the WT cells, and found expressions of *FOS*, *JUN* and *CD40* are among the top ones (Fig. S8C), that have been reported to regulate drug resistance and DNA repair in MM cells and other cancers²⁴⁻

Authors showed that majority of HP1 γ peaks were located within the non-coding regions (Fig. 6D) and they showed changes in the abundance of H3K36me3 and H3K9me3 marks at the promoter of those genes. Were the results confirmed in ATAC-seq dataset? How did genome-wide differential abundance of HP1 γ correlate with differential gene expression in RNA-seq data and eventually in resistance to bortezomib?

Response: For the first question, when looking at the ChIP-seq data in Figure 6D and ATAC-seq data in Figure 7B and 7D, we can find that similar with majority of HP1 γ peaks were located within the non-coding regions (~65%), majority of opened chromatin (~55%) were also seen on non-coding genes. This is reasonable, as we explained at the above question, HP1 γ is not a transcriptional factor but a H3K9me2/3 reader, therefore HP1 γ -binding area and HP1 γ -mediated chromatin accessibility are all genome wide. Since most of regions of human genes are non-coding sequences, and even the coding genes are mainly composed of introns, therefore these two data are correct.

Figure 6

Figure 7

As to how genome-wide differential abundance of HP1 γ correlates with differential gene expression and eventually in resistance to bortezomib, we speculate that changes of HP1 γ binding and chromatin accessibility at promoters of certain genes, will activate gene transcription and expression. Thus, we intersected the upregulated genes in RNA-seq and differently immunoprecipitated genes in ChIP-seq of BR MM cells, and screened 188 candidate genes (Fig 6D), then we double checked expressions of all these candidate genes in BR MM cells using qPCR, and finally determined *CD40*, *JUN* and *FOS* as key genes for chemoresistance. This is

our strategy, and wish it also answers the question above. We wish the reviewer would be satisfied with our explanation. Thanks!

Authors only selected three HP1 γ target genes due to their roles in DNA repair and drug resistance based on previous studies. It would have been nice if in a more unbiased approach, the critical genes would have been discovered and validated.

Response: Per your suggestion, we have re-analyzed the ChIP-seq and RNA-seq data, provided all intersected genes (Table S1), and validated expressions of some interested genes in the BR MM cells compared with WT parental cells (Fig. S 8C). Other genes may also play partial roles in promoting chemoresistance to PIs, but from our further functional validation, we confirm that CD40, JUN and FOS play major roles in this pathophysiological process.

Contribution of HP1 γ expression and stabilization in response to bortezomib have been tested in vitro and in vivo. Authors have shown that depletion of HP1 γ or its destabilization using HDAC inhibitors affect viability of tumor cells. Given the genome-wide impact of HDAC inhibition, authors did not exclude the off-target effects of those drugs on treated cells.

Response: This concern is relevant and insightful. Actually, HDAC inhibitors (HDACi), such as Panobinostat, have been approved for treatment of RRMM patients in the clinic (Garnock-Jones

KP., doi: 10.1007/s40265-015-0388-8). HDAC inhibitors exert anti-myeloma effects through multiple modes of action, the reported substrates including HDACs, p53, GATA1/2, STAT1/3, C/EBP α , HSP90, and modulating immune system (Yoichi Imai, doi: 10.3390/cancers11040475). To exclude the direct effect of HDACi on target genes, *CD40*, *JUN* and *FOS*, we treated endogenous HP1 γ expressing, or endogenous HP1 γ depleted BR-MM cells, and confirmed that HDACi only affects expressions of *CD40*, *JUN* and *FOS* when HP1 γ is in presence, thus we exclude the direct effect of HDACi on these target gene expressions. The new data were shown in the S Fig 8D:

D

And we updated the descriptions in the main text:

Firstly, we excluded the direct effect of HDACi on these genes, since HDACi only affected expressions of *CD40*, *JUN* and *FOS* when endogenous HP1 γ was in presence, but not in HP1 γ depletion BR-MM cells (Fig. S8D).

Some of the labelling of the figures and figure legends throughout this manuscript do not correspond to what is written in the main text. It was very confusing to read and understand what data is being described in each figure (no Supplementary figure 7, Fig. 4h, Fig. 3M and 3N). Quality of Supplementary figures was very low, and it was very hard to read the figures.

Response: We are very sorry for these careless mistakes. In the revised version, we have thoroughly checked the figures and corresponding description, and corrected all mistakes.

Line 279: MSC1 instead of MDC1.

Response: This error has been corrected. All authors are grateful for your careful review, and we are looking forward to your approval of this version.

Reviewer #2 (Remarks to the Author):

Overall, I praise the effort by Li et al to build on the constructive feedback by all three reviewers. I believe the manuscript improved, particularly in addressing the functional link between HP1gamma-linked molecular actuators (JUN, FOS, CD40) and drug resistance (e.g., Fig. S8H-J). I also appreciate the effort to add quantifications to the IF and Western Blots. That said, however, I still find issues with the overall rigor and mechanistic interpretation of the HP1gamma condensation data. As detailed below, I urge the authors to address these remaining major concerns thoroughly before publication:

Major-1: The interpretation and quantitative analysis of intracellular HP1gamma (and MDC1) puncta (e.g., Fig. 4C, Fi. S8H-J) are largely obscured by the lack of a proper definition of what they consider intracellular puncta. This issue is complex, and should be addressed/acknowledged, given that the aggregation patterns are so dissimilar across the range of cells and treatments. The quality of Fig. S8H-I images is too low for future readers to judge puncta. For example, I can barely tell the difference between MM1.5 +/- Rom for the MDC1 staining; yet quantifications look striking. Crucially, the new provided methods misses the mark in helping understand how they segment and count puncta. Moreover, quantifications usually show 3 data points, but I would expect to see many data points (wherein each data point corresponds to number of puncta per cell) for each condition (i.e., for each bar; please add data for all quantified cells).

Response: All authors thank the reviewer for the positive evaluation and consideration of our efforts on the revision, and we appreciate your further concerns and suggestions for rigor and integrity of our study.

For the first concern about proper definition of intracellular puncta, we actually refer to literatures published on EMBO and Nature (Wegmann S., doi: 10.15252/embj.201798049; Kilic S., doi: 10.15252/embj.2018101379; Geiger F., doi: 10.15252/embj.2021107711; Larson AG., doi: 10.1038/nature22822). We have added description of relevant puncta in the **supplementary method**, as well as the quantitative analysis of puncta by fluorescence intensity using software. Please check the new information:

In vivo puncta of phase separation was identified according to previous reports¹⁻⁴. Quantification of Immunofluorescence images about puncta count and relative puncta FI (fluorescence intensity) were analyzed using the Image J 1.46r software for all samples in each group. About puncta count analysis, we measured all images by finding maxima and fixing the prominence value. For relative puncta fluorescence intensity analysis, we measured all images after converting the image type, inverting, correcting the optical density, and fixing the measurement threshold in Image J software.

We also addressed the issue that the aggregation patterns are so dissimilar across the range of cells and treatments in the discussion section:

The acetylation and deacetylation of IDRs spatiotemporally regulate protein aggregation and impact membrane-less organelle formation *in vivo*³³. However, protein aggregation patterns are diverse across the range of cells and treatments, the recognition and regulation are still obscure.

As to the number of cells counted for quantifications, we enrolled 30 cells compared to 3 cells in the previous version, wherein each data point corresponds to number of puncta per cell. Please check the revised Fig 3G, S4M, 4A, 4C, S5D, S7H, S7I, S7J:

Revised 3G:

Revised S4M:

Revised 4A:

Revised 4C:

Revised S5D:

Revised S7H-J:

The authors didn't add these crucial data. The new Fig. S8H-J data fail to link MDC1 and HP1gamma at the puncta level. They should study/quantify colocalization of the MDC1 and Hp1 gamma signals.

Response: We are sorry for missing this crucial data. Per this request (actual S7 H-J, our mistake of the wrong order), we quantified the colocalization of the MDC1 and HP1 γ signals by recognizing yellow puncta, since yellow is the overlapped color of HP1 γ (green) and MDC1 (red). Please check the revised Fig S7H-J:

Major-3: Totally unclear how data in Fig. S7I (related to statement in lines 277-278) show a reduction in puncta upon HDAC inhibition (+Rom). It isn't obvious from the IF images and the quantifications are not labeled --if read as other unlabeled quantifications where the trends are obvious, the data would suggest increase in puncta upon Rom treatment. Other examples of bar quantification without proper labels: Fig. 4C.

Response: Thank for your careful review. We have corrected this mistake. Fig 4C labeling is correct, but we supplemented the labels of quantifications, as we have shown above.

Major-4: I remain concerned by the insistence (e.g., lines 375-376) on the model that HP1gamma condensates recruit MDC1. The authors never quantify the colocalization of MDC1 and HP1gamma. They also did not study MDC1 gamma recruitment to sites of DNA damage under HP1gamma KD --despite having the tools to do that. Curiously, the subsequent discussion in lines 381 to 388 describes a model that puts MDC1 upstream of HP1gamma, as their data do support. The underlying discrepancy needs to be corrected in favor of what their data show -- unless they add new data to suggest that HP1gamma indeed recruits MDC1.

Response: We agree with the reviewer at this point, that our current data can't support MDC1 is upstream of and recruits HP1 γ . Therefore, we modified the results strictly corresponding to what the data show:

Taken together, these data suggest that the MDC1-HP1 γ complex exert the effect of DNA repair in order to enhance the BTZ resistance of MM cells.

We propose a new ~~machinery~~ mechanism that HDAC1-mediated deacetylation improves the nuclear condensation of HP1 γ , consequentially reinforces the ~~recruitment of complex formation~~

with MDC1 to favor DNA repair and altering the chromosomal accessibility of genes governing MM cell survival.

The current study proposes that HP1 γ recruited by MDC1 promotes drug resistance of MM through DNA repair ~~through MDC1 recruitment~~.

~~Herein we reveal that MDC1 acts as a scaffold for HP1 γ and guides HP1 γ to recognize DNA damage sites, which~~ Thus, findings of the current study provide a new perspective for HP1 γ to recognize non-H3K9me3-dependent DNA damage sites in MM cells.

Major -5: regarding the contribution to clinical translation, the authors argue that their study helps "define [the] applicable population" for combination therapy. They should add clarity to this statement, namely, what is the applicable population based on the new findings?

Response: We appreciate this opinion from the reviewer. Based on our findings, the adaptive population is RRMM patients with high expression of HP1 γ , thus we can detect the expression of HP1 γ in bone marrow biopsy, and narrow the adaptive patients to acquire better treatment response. We modified the discussion as:

our study also provides new clue to clarify the applicable population, who should have high HP1 γ protein level in bone marrow biopsies after PI-based regimens, and thus contribute the development of precision medicine in MM management.

Minor:

- Fig S8 and S9 are mislabeled (should be s7 and s8)

Response: This error has been corrected.

- Fig. S8J, what is DMS? DMSO?

Response: Yes, it is DMSO. Thanks!

- Fig. 4I, note that the X axis label for "40" is misplaced

Response: This mistake has been corrected.

- Fig. 4H-I is presented as evidence that HDAC1-mediated deacetylation enhances LLPS, but the figure is mislabeled as "acetylation" for the treated group.

Response: We are sorry for this incorrect label. It has been corrected.

- What is the control in 4J? (vector?). The use of "vector" throughout the manuscript is often imprecise. If they mean "empty vector", then say so.

Response: Yes, we change all vectors to "empty vector".

- Lines 206-208. "Moreover, 206 in MM cells, when the HDAC1 was suppressed by its inhibitor Rom, or forcedly expressed, the nuclear condensation of HP1 γ was remarkably inhibited or promoted, respectively (Fig. 4C)." Fig. 4C shows OE, so missing SI citation.

Response: We apologize for this mistake, we actually only showed HDAC1 OE results. This result has been corrected as:

Moreover, in MM cells, when the HDAC1 was ~~suppressed by its inhibitor Rom, or forcedly expressed~~, the nuclear condensation of HP1 γ was remarkably ~~inhibited or promoted, respectively~~ (Fig. 4C).

-Line 210 "above proved that K5" should be "probed K5"

Response: This error has been corrected accordingly.

- Line 220-221 incorrectly states that the data for FL corresponds to HEK293 in vivo; when they seem to correspond to an in vitro assay --and the HEK data are for the IDR1 variants

Response: This is an incorrect description of the figure legend, but the figure is for FL proteins. Data for FL and IDR variants protein in vitro are shown in figure 4G and S 6B. We corrected the corresponding figure legend as you mentioned. Thanks!

-Line 279. Note "MSC1" puncta instead of "MDC1"

Response: This error has been corrected.

-Line 368: "new machinery" is not the proper term. New mechanism?

Response: Yes, mechanism is appropriate and we have replaced it. Thanks!

-Line 370: should be "which may reinforce". The authors do not provide convincing data to show that HP1gamma condensation recruits MDC1 to sites of DNA damage --they do provide data to show that MDC1 at sites of DNA damage is required for HP1gamma recruitment. I pointed this out before, and the authors largely corrected related statements, except here.

Response: We apologize for missing the modification of this interpretation. In the revised version we have removed any statement related to this point. Please check the above answers to major concern 4. Thanks!

-Lines 421: "trails" should be "trials". The authors should also cite the corresponding literature.

Response: We have corrected this verbal mistake and cite the reference No. 36. All authors greatly appreciate your insightful comments on our manuscript, and we are looking forward to your approval of this version.

REVIEWER COMMENTS

Reviewer #1 (Remarks to the Author):

no further comments

Reviewer #2 (Remarks to the Author):

Overall, I appreciate the efforts by the authors to add proper quantifications to the HP1g/MDC1 puncta data (30 cells vs 3 cells), as well as the new edits to the proposed model --to align it fully with the data. I also share an appreciation for the overall body of work advanced by the authors, and the potential clinical relevance of the findings to the treatment of multiple myeloma. Three comments that merit attention before publication:

Methods: The new description of the image processing approach remains insufficient for anyone to reasonably attempt reproducing their data. Adding citations is good but not sufficient. The authors should clearly explain their image processing criteria (e.g., puncta larger than, "fixing the prominence value" to XXX?). They often analyze puncta that cannot be easily (or at all) counted/seen by simple visual inspection of the images, so rigorous description of the image processing approach/criteria will be critical for future readers to make sense of the data.

Proposed model: The changes in language to avoid the issue of placing MDC1 upstream of HP1g (without supporting evidence) are OK, except for the text: "The current study proposes that HP1 γ recruited by MDC1 promotes drug resistance of MM through DNA repair". This statement still places MDC1 upstream of HP1 γ . Lines 188-189 also repeat this unsupported message: "Taken together, these data suggest that MDC1 recruits HP1 γ to exert the effect of DNA repair". The statement also appears in the abstract: "HP1 γ was recruited by the mediator of DNA damage checkpoint 1".

Figures: Fig. 4B shows a scale bar that indicates "5 μ m" --must be wrong considering the size of the nucleus.

RESPONSE TO REVIEWERS' COMMENTS

REVIEWER #2

Overall, I appreciate the efforts by the authors to add proper quantifications to the HP1g/MDC1 puncta data (30 cells vs 3 cells), as well as the new edits to the proposed model --to align it fully with the data. I also share an appreciation for the overall body of work advanced by the authors, and the potential clinical relevance of the findings to the treatment of multiple myeloma. Three comments that merit attention before publication:

Methods: The new description of the image processing approach remains insufficient for anyone to reasonably attempt reproducing their data. Adding citations is good but not sufficient. The authors should clearly explain their image processing criteria (e.g., puncta larger than, "fixing the prominence value" to XXX?). They often analyze puncta that cannot be easily (or at all) counted/seen by simple visual inspection of the images, so rigorous description of the image processing approach/criteria will be critical for future readers to make sense of the data.

Response: All authors greatly appreciated your positive evaluation and detailed suggestion for this version.

The concern about the definition of puncta is relevant and very important, especially for the purpose of reproducing our data. Per this request, we have added a detailed description for defining puncta in the **supplementary method**, as well as the quantitative analysis of puncta by fluorescence intensity using the software. Please check the new information:

Images for each channel (blue, green, and red) were exported as individual non-compressed TIFF files by the built-in FV10-ASW viewer software and individually analyzed by Photoshop. For image analysis, only the Brightness/Contrast of the entire

image was adjusted manually, and the parameters were kept the same for the same batch of images under the same channel (different channels may be adjusted differently). To preclude experimental variations, only puncta for HP1 γ or MDC1 in images obtained from the same experiments that have been applied with the same brightness/contrast adjustment were analyzed. For presentation, representative raw images were adjusted in brightness and contrast linearly and equally within the samples using ImageJ software. Image merging and cropping were done in Photoshop. Scale bar was applied to individual images before cropping. The puncta size for HP1 γ (wild type or overexpressed mutants) and MDC1 were analyzed by ImageJ software based on the standard procedures. Particle analysis was used in ImageJ to calculate the fluorescent intensity and area of puncta condensates in each sample, and we applied the script to all raw image sets in batch mode. In the script, a Watershed Separation and background correction was applied to each image, followed by particle analysis. We measured all images after converting the image type, correcting the optical density, adjusting and fixing the image threshold, separating particles using Watershed (Process-Binary-Watershed), and analyzing particles in ImageJ software (Please refer to more detailed procedure from Particle Analysis (imagej.net) and Particle Analysis - ImageJ). Each parameter was determined from particle analysis for endogenous HP1 γ in WT MM.1S cells. The exported data table was further summarized in Excel and GraphPad software. According to the sizes and pixels of the original figure, we calculated the sizes of particles. For HP1 γ or MDC1 puncta formed, we determined their size to be $1.057 \mu\text{m}^2$ (median in the area) with 20% and 95% percentiles to be $0.523 \mu\text{m}^2$ and $5.178 \mu\text{m}^2$, respectively. Hence, we counted those dots with area ranging $0.523 \mu\text{m}^2$ to $5.178 \mu\text{m}^2$. Similarly, we measured the co-location size for the yellow puncta formed by HP1 γ and MDC1, and defined the puncta with the same standard of 20% and 95% percentiles.

Proposed model: The changes in language to avoid the issue of placing MDC1 upstream of HP1 γ (without supporting evidence) are OK, except for the text: "The current study proposes that HP1 γ recruited by MDC1 promotes drug resistance of MM through DNA repair". This statement still places MDC1 upstream of HP1 γ . Lines 188-189 also

repeat this unsupported message: "Taken together, these data suggest that MDC1 recruits HP1 γ to exert the effect of DNA repair". The statement also appears in the abstract: "HP1 γ was recruited by the mediator of DNA damage checkpoint 1".

Response: We apologize for missing these corrections, and we modified these descriptions as:

HP1 γ ~~was recruited by~~ interacts with the mediator of DNA damage checkpoint 1 (MDC1) to induce DNA ~~damage~~ repair, and simultaneously the deacetylation modification and the interaction with MDC1 enhanced the nuclear condensation of HP1 γ protein and the chromatin accessibility of its target genes governing sensitivity to PIs, such as CD40, FOS and JUN.

Taken together, these data suggest that MDC1 ~~recruits~~ binds HP1 γ to exert the effect of DNA repair in order to enhance the BTZ resistance of MM cells.

The current study proposes that HP1 γ ~~recruited by~~ binding to MDC1 promotes drug resistance of MM through DNA repair.

Our study reveals that MDC1 plays a critical role in mediating the ~~recruitment~~ recognition of HP1 γ into DNA damage sites.

Figures: Fig. 4B shows a scale bar that indicates "5 μ m" --must be wrong considering the size of the nucleus.

Response: We are sorry for the wrong label. That should be 0.5 μ M not 5 μ M. Per this reminder, we also double checked other labels, and corrected other wrong labels of Fig 5H and 5F, which should be 10 μ M but not 20 μ M. Thank you very much for your careful review!

Figure 4B:

Figure 5F:

Figure 5H:

REVIEWERS' COMMENTS

Reviewer #2 (Remarks to the Author):

The revised manuscript addressed my concerns.